# Retrotranspositional landscape of Asian rice revealed by 3000 genomes

Marie-Christine Carpentier[1], Ernandes Manfroi[2], Fu-Jin Wei[3,4], Hshin-Ping Wu[3], Eric Lasserre[1], Christel Llauro[1], Emilie Debladis[1], Roland Akakpo[1], Yue-Ie Hsing[3] & Olivier Panaud [1,5]

The recent release of genomic sequences for 3000 rice varieties provides access to the genetic diversity at species level for this crop. We take advantage of this resource to unravel some features of the retrotranspositional landscape of rice. We develop software TRACK-POSON specifically for the detection of transposable elements insertion polymorphisms (TIPs) from large datasets. We apply this tool to 32 families of retrotransposons and identify more than 50,000 TIPs in the 3000 rice genomes. Most polymorphisms are found at very low frequency, suggesting that they may have occurred recently in agro. A genome-wide association study shows that these activations in rice may be triggered by external stimuli, rather than by the alteration of genetic factors involved in transposable element silencing pathways. Finally, the TIPs dataset is used to trace the origin of rice domestication. Our results suggest that rice originated from three distinct domestication events.

[1] Laboratoire Génome et Développement des Plantes, UMR CNRS/UPVD 5096, Université de Perpignan Via Domitia, 52 Avenue Paul Alduy., 66860 Perpignan Cedex, France. [2] Faculdade de Agronomia, Universidade Federal do Rio Grande do Sul, Porto Alegre, RS 90040-060, Brazil. [3] Institute of Plant and Microbial Biology, Academia Sinica, 128, Section 2, Yien-chu-yuan Road, Nankang 115 Taipei, Taiwan. [4] Department of Forest Molecular Genetics and Biotechnology, Forestry and Forest Products Research Institute, 1 Matsunosato, Tsukuba 305-8687 Ibaraki, Japan. [5] Institut Universitaire de France, 1 rue Descartes, 75231 Paris Cedex 05, France. Correspondence and requests for materials should be addressed to O.P. (email: panaud@univ-perp.fr)

One of the major discoveries of the last decades of genomic research is that genes are outnumbered by transposable elements (TEs) in most eukaryotic genomes[1,2]. In flowering plants, the distribution of genome size among 7500 angiosperm species indeed shows that 99% of the species have a genome larger than 200 Mbp/1 C (i.e., at least twice the size of their gene space, ~100 Mbp on average)[3].

TEs are very diverse both in terms of structure and modes of transposition[4], but they all can be mobilized and amplified within the genomes, which explains their propensity to densely populate eukaryotic chromosomes[5]. TEs have long been considered useless or even deleterious, but these views have recently been challenged. Experimental evidence have shown that they could be beneficial to organisms and have, in some cases, been domesticated by their host genome in various eukaryotic lineages to create biological novelty[6,7]. With the advent of new sequencing technologies and the availability of genomic resources for many organisms (and even populations), the molecular mechanisms involved in this process have begun to be unraveled and show that both the regulatory sequences and the proteins encoded by TEs can be exapted[8,9]. In a shorter time scale, TEs have been shown to play a role in adaptation in natural populations[10] and in crops[11,12].

Several aspects of TE dynamics at the population level remain unclear. On the one hand, comparative genomic studies in various lineages show that TEs contribute significantly to genome diversification, suggesting that TE insertion polymorphisms (TIPs) could be frequent enough in natural populations to serve as a source of adaptive variation[13]. On the other hand, recent advances in epigenetics clearly show that transposition is strictly controlled in planta by several transcriptional and post-transcriptional pathways[14], which raises the questions of the conditions of transposition activation *in natura* and of the actual dynamics of transposition in natural populations.

The exhaustive characterization of TIPs in a given gene pool requires genomic data for a comprehensive sample of individuals and at least one good-quality reference genome sequence from which TEs have been well characterized. These resources are available for a few model species. Rice (*Oryza sativa*), is well suited for such study with one high quality, physical map-based genome assembly[15], from which TEs have been annotated and curated[16,17]. More importantly, with a genome size of 430 Mbp, rice is close to the most representative plant genome in terms of size (and thus TE content), as evidenced by the modal value of the distribution of 7500 angiosperm lineages, which is ~500–600 Mbp[3]. Rice genome indeed harbors hundreds of TE familes of both class I and class II types[16,18,19]. Long Terminal Repeats (LTR)-retrotransposons constitute the largest part of rice mobilome in terms of percentage of genome they represent[20]. In a previous study, we showed that retrotranspositional activity in *Oryza* genus was at the origin of significant variations in genome size among diploid species, suggesting that this particular type of TEs plays a major role in transposition-driven genome dynamics in this lineage[21]. Rice genome harbors ~300 families of LTR-retrotransposons, belonging to either *Gypsy* or *Copia* superfamilies[16]. These families differ in copy number, ranging from singletons to large families (e.g., over 100 complete copies for *Hopi* and *Houba* families). In addition to the high-quality map-based genome assembly of the Nipponbare variety, three other high-quality PacBio-based assemblies are publicly available[22] and the raw Illumina-based genome sequences of 3000 accessions have been released[23]. This offers the opportunity to study genome dynamics at intra-specific level and at an unprecedented scale, although this requires the development of bioinformatic tools that are suitable for the analysis of very large datasets.

Conceptually, the detection of TIPs from populations based on Illumina data is straightforward, although prone to high false discovery rate (FDR) for large genomes (several 100 Mbp) due to the small reads size[24]. The methods commonly used are paired-end mapping (PEM) and split reads. For the first method, all paired reads are mapped onto a reference genome and discordant pairs (i.e., both reads from the same "amplicon" mapping at different locations) should correspond to structural variants and declared as TIPs if one end matches a known TE. The second method consists in identifying reads for which one part maps onto the reference genome sequence while the other matches a TE[25–27]. The main limitation of these approaches (besides FDR) is that the mapping step of all paired reads onto a reference genome is computationally intensive.

Therefore, we developed TRACKPOSON software for the efficient detection of TIPs of known TEs in large datasets. Using this new tool, we successfully characterized TIPs among 3000 rice genomes for 32 retrotransposons families. We chose this sample as being representative of the 300 families found in the rice genome in terms of superfamilies, copy number and transpositional activity[16,28–31], thus unraveling the retrotranspositional landscape of a plant genome at the species level. Using this data, we tentatively looked for genetic factors that may trigger transposition in agro in rice and conclude that for most TE families, such activation likely originated from external stimuli, rather than from a mutation within a genetic factor involved in the control of transposition.

The origin of rice domestication has been strongly debated in the past decades. Many studies of the genetic diversity of Asian rice clearly show a diphyletic origin of the crop, thus suggesting two independent domestications[32,33]. However, the cloning and characterization of some key domestication loci, such as shattering locus *Sh4*[34], showed that all rice types harbor the same domesticated allele, which suggests a single domestication event and that the differenciation of rice into the two major *Indica* and *Japonica* types may result from introgressions between this first domesticate and wild relatives in southern Himalayas[35,36]. Alternatively, one could hypothesize that there were indeed two distinct domestications, followed by some introgressions between the two domesticated gene pools, as the result of human selection for the best cultivated phenotype[37]. Finally, the domesticated alleles could have been present in the populations of the wild progenitor of rice long before the split of the two gene pools that gave rise to the domesticated forms and been selected for at least two times independently[38]. Here, exploiting TIPs as genomic paleontological records, we show that *Indica, Japonica*, and *Aus/ Boro* groups originate from wild relatives that diverged long before upper neolithic, which supports the hypothesis that they originated from multiple domestication events.

## Results

**TRACKPOSON is a tool for the detection of TIPs in gene pools.** The new strategy we developed consists in first mapping all reads of a given accession onto each TE family represented by a single consensus sequence (as opposed to mapping them onto the complete genome, as in the case of conventional PEM procedures) and then mapping the unmapped paired reads onto the rice reference genome, split into 10 kb windows (Fig. 1). In this regard, TRACKPOSON is not designed for a full characterization of all structural variations, but rather as a fast tool to unravel the transpositional activity of known TE families from very large genomic dataset. We first tested TRACKPOSON on a rice mutant for which TIPs had been previously characterized by using our previous PEM-based software[39] and wet-lab validated by PCR amplification and sequencing (Supplementary Figure 1). All TE

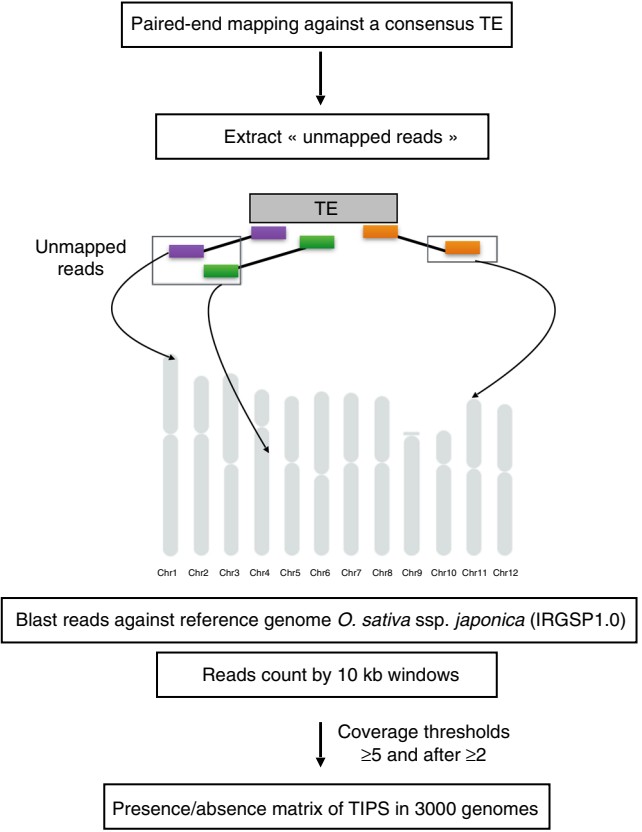

Paired-end mapping against a consensus TE

Extract « unmapped reads »

TE

Unmapped reads

Chr1 Chr2 Chr3 Chr4 Chr5 Chr6 Chr7 Chr8 Chr9 Chr10 Chr11 Chr12

Blast reads against reference genome *O. sativa* ssp. *japonica* (IRGSP1.0)

Reads count by 10 kb windows

Coverage thresholds ≥5 and after ≥2

Presence/absence matrix of TIPS in 3000 genomes

**Fig. 1** TRACKPOSON method. Schematic representation of the pipeline

insertions were detected, which suggests that TRACKPOSON is a robust detection procedure with high sensitivity, given that sufficient genome coverage has been achieved (the mutant was sequenced at 30× depth). We should emphasize that the genome coverage of the 3000 rice accessions is 11.6× on average, with some as low as 7× (the first quartile being <9.4×). We therefore had to adapt our method to improve its sensitivity (low false-negative rate). First, only events supported by at least five paired reads were declared as TIPs, thereby opening the corresponding 10 kb window for the rice variety for which the polymorphism was identified. A second step consisting of another complete scan of the 3000 rice varieties, albeit with a detection threshold of 2 paired reads, was performed, and only for the 10 kb windows previously opened (supported by 5 paired reads). This considerably improved the sensitivity of detection (Supplementary Figure 2), with little risk that it increased the false-positive detection rate because the probability that two chimeric amplicons match the exact same 10 kb window as the one previously detected in another accession should be very low.

In order to check for the performance of this method, we resequenced one rice accession (Mutha Samba::IRGC 49924–2) that was originally sequenced in the 3000 genome collection at a 8.3× depth, i.e., among the lowest coverage in the dataset. We used Nanopore long-read sequencing technology (Minion device). One flowcell was used and produced 4.77 Gbp of sequence (11× genome coverage). We manually checked for the presence of TIPs of *Hopi*, *Tos17*, and *Karma*, from the reads thus generated[40]. We chose these three families because they are representative of the diversity of the retrotransposons found in the reference rice genome, i.e., a low (*Tos17*) and a high (*Hopi*) copy-number LTR-retrotransposon families and a moderately repeated LINE family (*Karma*). We should however emphasize

that TRACKPOSON could only unravel the transpositional activity of TE families identified from the reference genome sequence and therefore that we could not test the efficiency of this software on a retrotransposon family with genomic features that are very different from *Hopi*, *Tos17*, or *Karma*, in case such family exists in one of the 3000 rice genomes represented in the dataset. TRACKPOSON identified 501, 8, and 9 insertions of *Hopi*, *Tos17*, and *Karma*, in Mutha Samba cultivar, respectively. Hundred percent of the insertions of both *Tos17* and *Karma* identified by TRACKPOSON were validated. In the case of *Hopi*, 473 insertions were validated. These results show that the specificity of TRACKPOSON is ~94.5%. The rate of false negatives (sensitivity) was also estimated. While there was a 100% overlap between the insertions detected by TRACKPOSON and those detected using Nanopore reads for both *Tos17* and *Karma*, which is indicative of 100% sensitivity for these two families, we detected a total of of 581 *Hopi* insertions from the Nanopore dataset, which corresponds to a drop of sensitivity to 81% for this family. However, the mappability of a complex genome such as that of rice with small reads obtained on Illumina platforms is expected to not be 100% because of the presence of repeats that impede unambiguous mapping at unique sites[41]. In the case of rice, the mappability of 100 bp windows at a $1e^{-20}$ threshold (i.e., that used for the second step of the detection, see Methods section) was estimated to be 63.5%, on average. This estimation fits with previous estimates of the repeat content of *Japonica* rice genome (i.e., ~40%[15]). We determined the mappability of the regions where we detected *Hopi* insertions with Nanopore reads. We found an average of 58% for all insertions. However, the insertions found only in the nanopore data (and not detected by TRACKPOSON) have a significant lower mappability of 42%. This may explain the lower sensitivity of our software for this particular family.

**Retrotranspositional landscape of Asian rice.** TRACKPOSON was used to detect TIPs for 32 retrotransposon families in the 3000 rice genomes dataset. The number of complete elements for these families in the reference rice genome is given in Table 1. We chose a sample of families that are representative of the diversity found in this accession in terms of number of repeats, from very moderately (e.g., *Tos17* with two complete copies in the reference genome) to highly repeated families (e.g., *Houba* with 150 complete copies). Therefore, we consider that the features revealed by this sample of retrotransposons should be representative of the complete retrotranspositional landscape of rice genome shaped by the whole 300 LTR-retrotransposon families[16]. In total, we identified 53,262 and 47,007 TIPs for the 3000 genomes and the 1067 traditional varieties included in the dataset, respectively (Table 1). Each family showed some variation of copy number among the 3000 rice accessions (Fig. 2a). Overall, the total number of insertions when all families were considered varied from 3324 to 12,380 with an average of 6225. No rice accession was found to be an outlier in terms of LTR-retrotransposon content.

We then analyzed the map position of all the TIPs. As mentioned above, we first determined the mappability of the rice genome in order to avoid bias in the interpretation of our results. As shown in Fig. 3, the mappability of TIPs decreased in pericentromeric regions, known to be repeat-rich. The mapping of all TIPs on the 12 rice pseudomolecules however shows an insertion bias in these regions, especially for *Gypsy* elements (Fig. 3), which confirms previous studies[42,43]. However, the mapping data shows that no region of the genome is devoid of TIPs, which suggests that retrotransposition contributes to genome diversity in all chromosomes regardless their position.

**Table 1 TRACKPOSON results for 32 transposable elements families in 3000 rice genomes[a]**

| TE family | Families | Total TE insertions number | Total TE insertions number in traditional rice | Mean insertion in *japonica* varieties | Mean insertion in *indica* varieties | Distance from gene (kb) | Activity |
|---|---|---|---|---|---|---|---|
| *Poprice* | *Copia* | 2324 | 1678 | 98.3 ± 12.1[b] | 91.6 ± 14.0[b] | 22.5 ± 31.4[b] | Recent |
| *Tos17* | *Copia* | 181 | 121 | 7.6 ± 3.6 | 7.9 ± 2.9 | 21.1 ± 35.8 | Recent |
| *Houba* | *Copia* | 5976 | 5112 | 523.8 ± 65.1 | 388.8 ± 57.0 | 22.6 ± 32.0 | Recent |
| *Fam89_osr7* | *Copia* | 587 | 425 | 47.4 ± 8.7 | 32.0 ± 7.0 | 23.4 ± 33.3 | Recent |
| *Fam35-fam36* | *Copia* | 1756 | 1678 | 669.0 ± 63.7 | 637.2 ± 77.0 | 22.9 ± 32.2 | Old |
| *Fam67_echidne* | *Copia* | 438 | 371 | 105.8 ± 20.5 | 102.2 ± 24.4 | 23.4 ± 33.4 | Recent |
| *Rn304* | *Copia* | 52 | 30 | 2.2 ± 1.5 | 1.9 ± 1.2 | 24.8 ± 36.9 | Continuous |
| *Scaff6* | *Copia* | 29 | 23 | 1.7 ± 0.5 | 1.9 ± 0.9 | 27.9 ± 41.0 | — |
| *Lullaby* | *Copia* | 153 | 92 | 5.6 ± 2.1 | 3.1 ± 1.8 | 26.8 ± 37.4 | Recent |
| *Fam51_osr4* | *Copia* | 1788 | 1328 | 104.6 ± 11.0 | 100.0 ± 12.2 | 22.6 ± 31.6 | Recent |
| *Fam90* | *Copia* | 285 | 166 | 17.9 ± 2.9 | 16.6 ± 4.1 | 24.4 ± 32.5 | Recent |
| *Fam93_ors14* | *Copia* | 2194 | 1521 | 22.9 ± 13.5 | 56.4 ± 12.9 | 22.0 ± 31.8 | Recent |
| *Fam98_rn81* | *Copia* | 153 | 123 | 20.6 ± 5.1 | 10.8 ± 4.0 | 26.4 ± 35.1 | Recent |
| *Hopi* | *Gypsy* | 5152 | 5027 | 695.7 ± 179.5 | 701.3 ± 215.0 | 23.5 ± 33.7 | Continuous |
| *Dagul* | *Gypsy* | 2924 | 2742 | 571.8 ± 67.9 | 527.8 ± 75.6 | 22.6 ± 31.9 | Recent |
| *Fam17_Rn215_125* | *Gypsy* | 7096 | 7006 | 575.9 ± 175.5 | 930.1 ± 319 | 23.8 ± 34.6 | Continuous |
| *Fam80_rire7* | *Gypsy* | 382 | 338 | 66.5 ± 12.8 | 69.9 ± 14.2 | 23.4 ± 32.8 | Continuous |
| *Dasheng* | *Gypsy* | 5723 | 4806 | 586.4 ± 60.5 | 486.7 ± 61.1 | 22.8 ± 32.3 | Recent |
| *Rire2* | *Gypsy* | 3061 | 2785 | 304.6 ± 53.0 | 295.9 ± 70.3 | 22.8 ± 32.2 | Recent |
| *Fam81-fam82* | *Gypsy* | 2758 | 2558 | 474.2 ± 74.3 | 463.3 ± 94.3 | 23.6 ± 34.5 | Continuous |
| *Fam31_osr37* | *Gypsy* | 3368 | 2797 | 393.1 ± 53.4 | 430.7 ± 57.0 | 23.4 ± 33.2 | Recent |
| *Fam49_osr29* | *Gypsy* | 968 | 904 | 304.4 ± 36.0 | 284.0 ± 44.6 | 23.5 ± 33.6 | Old |
| *Fam124_rn208* | *Gypsy* | 112 | 68 | 11.02 ± 1.9 | 10.47 ± 1.8 | 26.3 ± 35.8 | Continuous |
| *Fam108* | *Gypsy* | 594 | 570 | 201.5 ± 31.3 | 188.1 ± 36.6 | 26.2 ± 37.6 | Continuous |
| *Fam106* | *Gypsy* | 213 | 148 | 15.8 ± 3.7 | 9.3 ± 3.7 | 23.7 ± 32.7 | Recent |
| *Fam86* | *Gypsy* | 506 | 450 | 202.1 ± 16.7 | 192.0 ± 20.8 | 23.9 ± 33.5 | Continuous |
| *Fam79_rn206* | *Gypsy* | 601 | 461 | 125.8 ± 10.8 | 120.9 ± 13.3 | 23.5 ± 32.4 | Recent |
| *Rn60* | *Gypsy* | 16 | 15 | 1.3 ± 1.4 | 1.2 ± 0.9 | 25.6 ± 38.7 | — |
| *Scaff3* | *Gypsy* | 56 | 28 | 1.4 ± 1.1 | 1.9 ± 1.3 | 32.8 ± 48.1 | Recent |
| *Scaff5* | *Gypsy* | 19 | 1 | 1 ± 0 | 0.7 ± 0.5 | 27.0 ± 42.6 | — |
| *Rire3* | *Gypsy* | 3719 | 3576 | 458 ± 147.0 | 408.2 ± 152.3 | 23.9 ± 34.9 | Continuous |
| *Karma* | *LINE* | 78 | 59 | 3.7 ± 1.4 | 4.2 ± 2.2 | 25.2 ± 36.7 | Recent |
| 32 families | | 53,262 | 47,007 | | | | |

[a]The transpositional history of each TE families was based on the histogram in Fig. 2
[b]Numbers after "±" denote standard deviation

Furthermore, the position of the TIPs in relation to the closest gene differed significantly between *Gypsy* and *Copia* elements ($t$ test, $p$ value $<1.6^{e-7}$), with the *Copia* elements insertions being closer to genes than the *Gypsy* elements (Table 1; method described in Supplementary Table 3).

The level of polymorphism generated by the 32 retrotransposon families was assessed by using the frequency of insertions found in the 1067 traditional varieties (Fig. 2). Surprisingly, a large portion of TIPs are specific to only one variety (Fig. 2c). The sensitivity of TRACKPOSON is high but not 100% for highly repeated families. We therefore may not exclude the possibility that some insertions may have been missed in some accessions, but this may certainly not change the L-shape distribution observed in Fig. 2c. Therefore, our results strongly suggest that transposition may have occurred in agro, after domestication. Moreover, these low-frequency TIPs were found in all varietal groups, regardless of their geographical origin, which further suggests that transposition is triggered in various agro-environments. However, the 32 families did not contribute to genome diversification in the same fashion (Fig. 2b): some families (e.g., *Houba*) exhibit only very low-frequency insertions (L-shaped distribution of the number of accessions sharing the insertions), which suggests a recent transpositional activity or a segregation of ancient polymorphisms via lineage sorting. However, some, like *Hopi*, exhibited insertions at frequencies ranging from ~0.001 (insertions found in only one variety) to ~1 (ancestral insertions found in all traditional varieties). This finding suggests that such families have undergone transposition continuously since domestication and that the low frequency of *Houba* insertions is likely not due to lineage sorting, but to recent activity. Finally, for a few families (e.g., *Dasheng*), most insertions were found at high frequencies, which suggests more continuous activity.

The frequency of each TIP in the 1067 traditional varieties, together with its map location clearly shows that the TIPs found at high frequency are mostly pericentromeric (Fig. 3). Such high frequency TIPs could be considered as old insertions (because they are shared by many varieties) as opposed to the ones found at low frequency. The higher TE density usually found in pericentromeric regions in plant genomes could thus result from retention of TEs in addition to an insertional bias in these regions for *Gypsy* elements.

**No genetic factor for transposition activation in rice.** By showing that retrotransposition contributes to genomic diversification in rice and to such a large extent, we confirm that it is a major force driving genome evolution in this crop and likely in plants in general. However, in rice, like in the other model species *Arabidopsis thaliana*, transposition is strictly controlled by several epigenetic pathways[44], which raises the question of whether transposition is triggered in agro by mutations in key genes

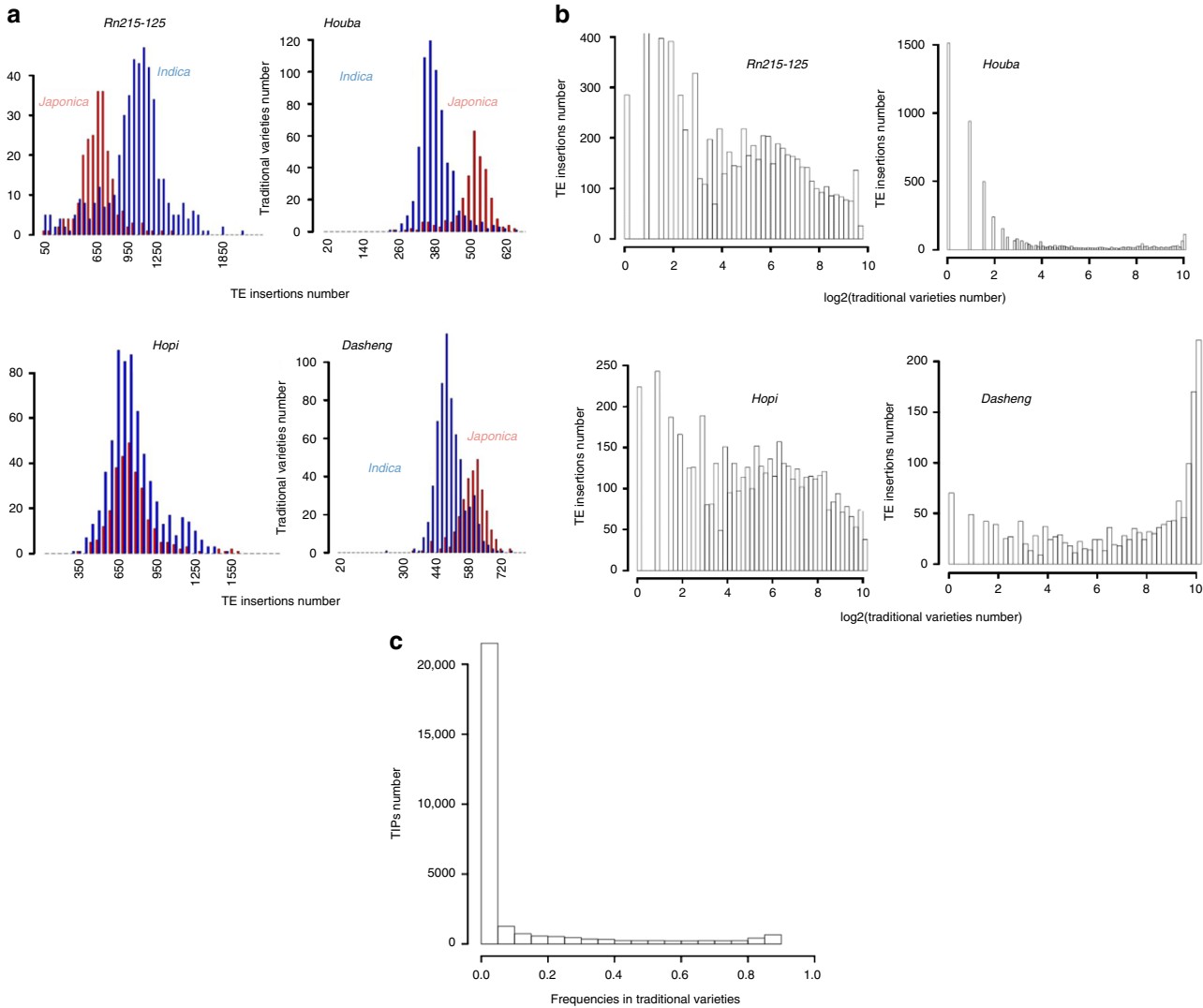

**Fig. 2** History of TE families. **a** Distribution of the number of TE insertions in traditional varieties. *x* axis represents the number of TE insertions for each TE family. *Indica* and *Japonica* varieties are shown in blue and red, respectively. **b** Distribution of traditional varieties by TE insertion. The *x* axis represents the number of rice traditional varieties in log2 scale and *y* axis the number of TE insertions. For all the TE families, the peak on the left corresponds to TE insertions present at very low frequencies in rice varieties, which suggests recent insertions. **c** Distribution of frequencies of TIPs in traditional varieties. *x* axis shows the frequency of TE insertion in traditional varieties and *y* axis shows the frequencies for all TE insertion in all 32 families. Most TIPs are present with low frequencies (<0.01), i.e., TE insertions are only in one or two rice varieties

involved in these pathways in cultivated populations or, alternatively, by external stimuli such as biotic or abiotic stress that may modify the epigenetic landscape of the genome, mimicking the impediment of these pathways. We tested both hypotheses by using a genome-wide association study (GWAS), similar to that of Quadrana et al.[45]. Using the single-nucleotide polymorphism (SNP) dataset generated in the framework of the 3000 rice genome project[46], we sought associations between these markers and the number of copies (taken here as a phenotype) for each retrotransposon family (Fig. 4 and Supplementary Figures 3–14). Significant association peaks were found for the 32 families, although the clearest results were obtained for the 12 less repeated families (e.g., *Tos17*, Fig. 4a). For the others, e.g. *Rire2* or *Rire3*, the peaks were too numerous. For the 12 less repeated families, a total of 26 significant peaks were found (Supplementary Figures 3–14). Twenty fell into a region with a copy of the element identified using our TRACKPOSON software. The remaining five peaks (*Fam86x2*, *Fam89*, *Fam124*, *Karma*, and *Houba*) did not fall within a locus harboring a copy of the retrotransposon.

However, the mappability in these regions was <60% because of the presence of repeats, which suggests that some insertions may not have been detected with TRACKPOSON.

That a majority of association peaks overlapped with a TE insertion suggests that the copy number of a given family depends on the presence of an insertion of a member of the family at a site where it can be activated in planta. We should stress out that such insertion may be distinct from the previously characterized active copies of well-known families, like in the case of *Tos17* and *Karma*: Nipponbare genome harbors two copies of *Tos17*, one on chromosome 10 (position 15.4 Mbp) and the other on chromosome 7 (position 26.7 Mbp). It was previously shown that the latter was the one activated during callus culture[29] and therefore, it is often referred to as the active copy of the family. The peak that we identified on chromosome 7 is located at ~20 Mbp (Fig. 4a) and thus does not overlap with that copy. However, we identified another *Tos17* insertion, shared by 404 varieties (366 *Indica*s and 17 *Japonica*s), in the region of the peak. This suggests that there is a copy of *Tos17* found in rice germplasm,

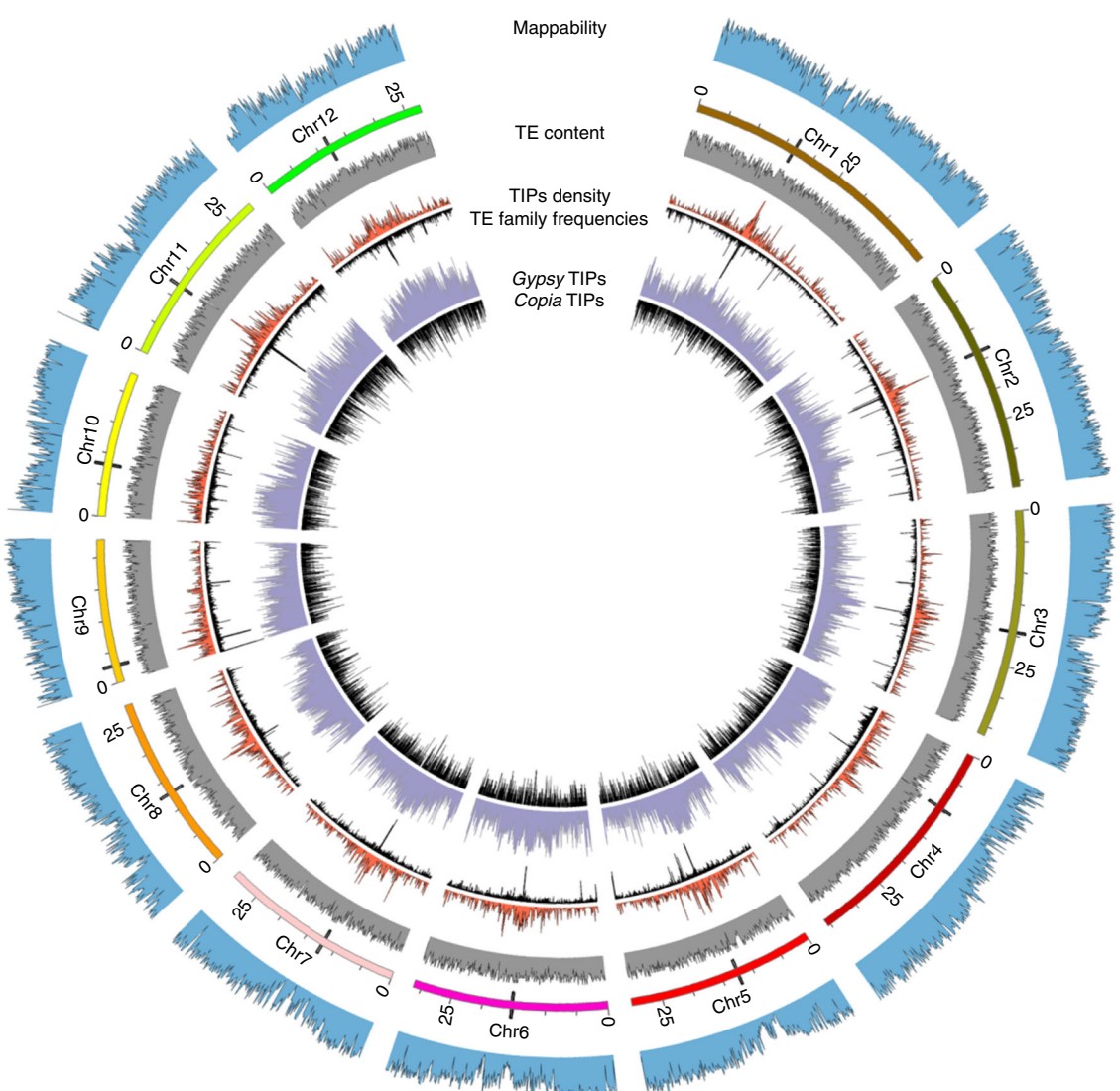

**Fig. 3** Circos representation. From outside to inside: the first circle corresponds to the mappability of the 12 rice chromosomes (see Methods). The second circle represents the 12 chromosomes of rice genome. For each chromosome, the black tips correspond to the centromere. The third circle represents the TE content: all TE insertions present in rice annotation file. The fourth circle (in red) corresponds to the mapping of the TIPs for all the 32 TE families and opposed, below, in black, the fifth circle shows the number of different TE families for each TE insertion polymorphisms (TIPs), (i.e., the scale is from 1 to 32). The sixth circle represents the distribution of TE insertions per LTR-retrotransposon type: *Gypsy* (purple) or *Copia* (black)

which may be more active than that of Nipponbare on chromosome 7, or at least which may be active in agro since its presence is correlated with an increase in copy number of the element in rice germplasm. The conditions of its transpositional activation remain however to be elucidated. In the case of the LINE *Karma*, Nipponbare genome harbors only one complete copy located on chromosome 11 (at ~27 Mbp). That particular insertion is found in 1558 rice lines (among which 821 from *Indica* type and 568 from *Japonica* type). It also harbors an inactive truncated copy (often found with LINEs) on chromosome 5 (at ~13.2 Mbp). *Karma* was identified as transpositionally active in rice calli[30], like for *Tos17*. The peak that we obtained in our GWAS is located on chromosome 7 (Fig. 4b). It overlaps with an insertion of *Karma*, shared by 697 lines (among which 469 from *Indica* type and 62 from *Japonica* type). It therefore appears that the most active *Karma* copy is not the one previously identified in Nipponbare, similar to what observed for *Tos17*.

The gene annotation of the 26 regions for which a significant association was found did not reveal the presence of any genetic factors known to be involved in the control of transposition[14],

unlike what was is found in *Arabidopsis*[45]. This result, together with the fact that most peaks overlap with a TE insertion, suggests that one cause of transposition activation in agro is the presence of an active copy of the element, rather than an alteration of a genetic factor controlling a cellular pathway for transposition control (i.e., epigenetic silencing). In addition, the presence of such active copy may not be sufficient, as transposition has been shown to be triggered in particular physiological states, such as biotic or abiotic stress in plants that may modify the methylation status of TEs[47]. In the case of rice, the exact nature of such external stimuli and the dynamics of the plant's response to it remain to be elucidated.

**Cultivated rice may originate from distinct domestication events**. We first used the TIPs data for principal coordinate analysis of the 3000 rice accessions, and showed that TE insertions, like SNPs, clearly discriminate *Indica* and *Japonica* varieties, while the *Aus/Boro* group appeared to be more similar to the *Indica* group (Fig. 5a). We then tentatively dated the origin of

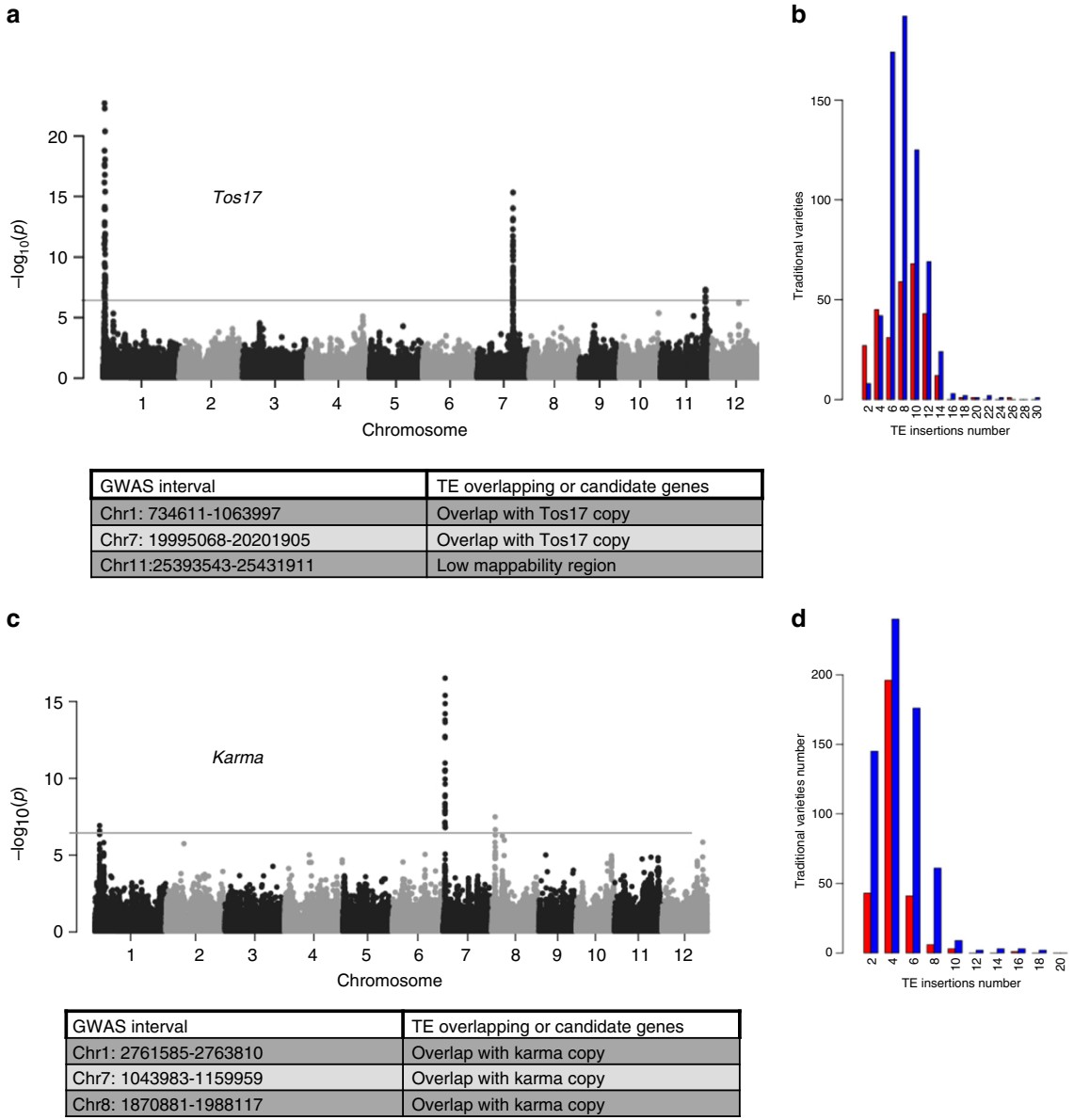

**Fig. 4** GWAS analysis of *Tos17* and *Karma* TE families. GWAS analysis for *Tos17* (**a**) and *Karma* (**b**) TE families. Manhattan plot represents the log10 *p* value for each association between the single-nucleotide polymorhisms and the TE insertion (see Methods). TE insertion-SNP association *p* value >6 are significative. **c**, **d** The TE insertion distribution in rice traditional varieties (Fig. 3) reflects a polymorphism in TE copy number along rice varieties, considered a quantitative phenotype

the three varietal groups *Japonica*, *Indica*, and *Aus/Boro* using a genomic paleontology approach[32]. For this, we first identified all insertions of full elements in Nipponbare (for *Japonica*), IR8 (for *Indica*), and N22 (for *Aus/Boro*) high-quality genome assemblies[22] for the nine TE families showing the highest number of TIPs, i.e., *Dagul*, *Dasheng*, *Hopi*, *Houba*, *Osr37*, *Rire2*, *Rire3*, *RN215*, and *Poprice* (see Methods section for the details of the procedure). These TIPs were then classified into seven distinct categories: (1), (2), and (3): *Indica*-, *Japonica*-, and *Aus/Boro*-specific insertions, respectively; (4) ancestral insertions present in all traditional varieties, regardless of their varietal group; (5) insertions that are common between *Japonica* and *Indica* but not present in *Aus/Boro*, (6) insertions that are common between *Indica* and *Aus/Boro*, but absent from *Japonica*, and (7) insertions that are common between *Japonica* and *Aus* but absent from *Indica*. As previously mentioned, most TIPs are found at low

frequency and therefore the insertions used for this analysis represented a small fraction of all TIPs (Fig. 6). Each TIP thus identified was dated using the method of SanMiguel et al.[48] with a molecular clock of $1.3 \times 10^{-8}$ substitution/site/year[49]. In total, we successfully dated 1476 TIPs from the seven categories (Fig. 6a). As expected, the insertions of the fourth category (i.e., common among all varieties) are more ancient than those belonging to the other categories (*Indica*- and *Japonica*-specific, respectively), which illustrates a split of a common wild ancestor into three distinct gene pools. The distribution medians for *Indica*, *Japonica*, and *Aus/Boro*-specific TE insertions are 99.4% identity (~230,000 years ago), 99.2% identity (~310,000 years ago), and 99% identity (~380,000 years ago), respectively. These values represent a peak of transpositional activity after the split of the lineages that gave rise to the three cultivated types and could therefore be used to estimate the lower limit of the time of divergence between the

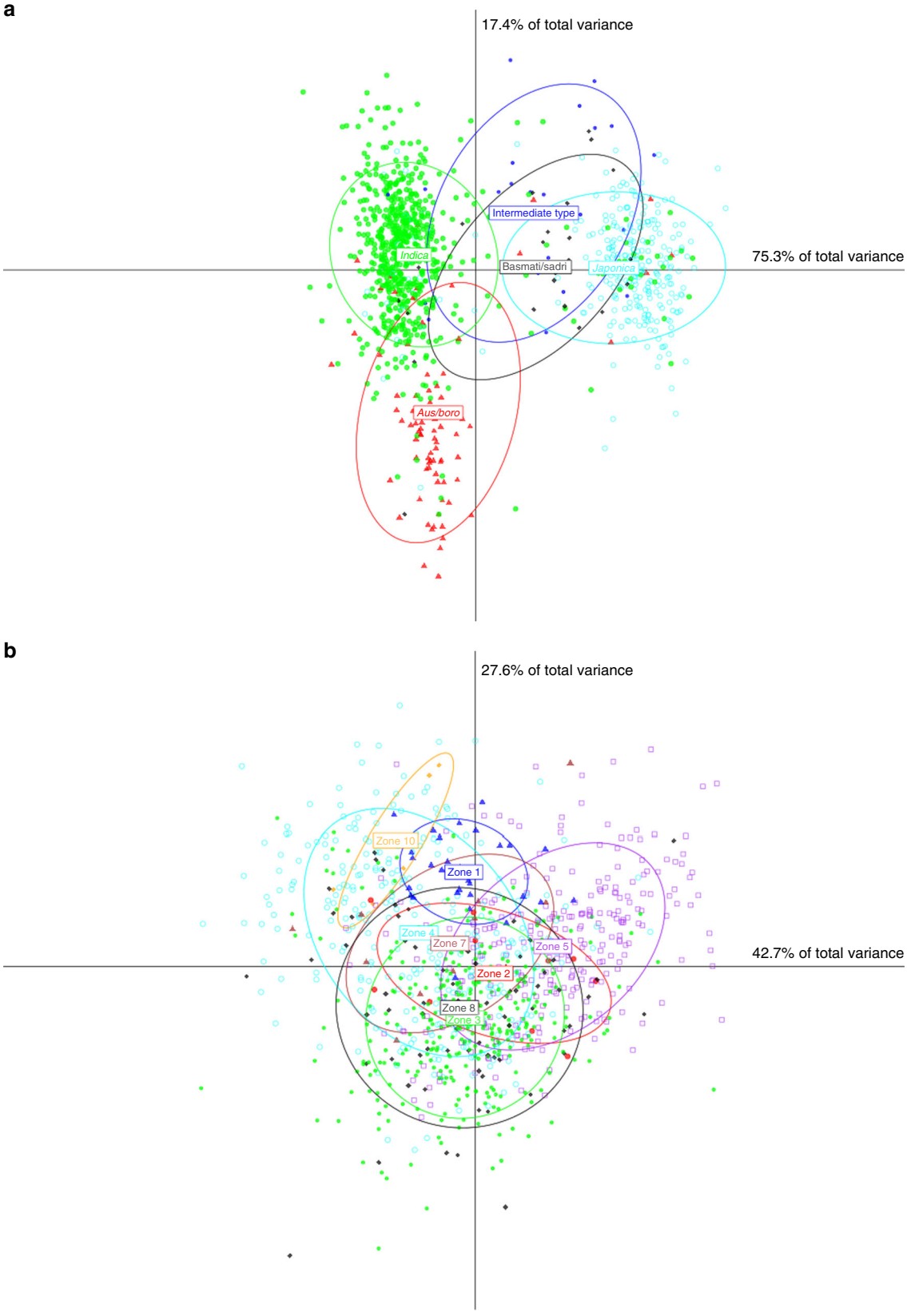

**Fig. 5** Discriminant analysis of the 3000 genomes using TIPs. Discriminant analysis of principal components was performed using TE insertions for all the 32 TE families as a function of varietal groups in green for the *Indica* varieties and in blue for the *Japonica* varieties (**a**) or in function of geographical zones (**b**). Zone 1 (blue): Japan and Korea; zone 2 (red): China; zone 3 (green): South-East Asia—Thailand, Laos, Malaysia, Cambodia, and Myanmar; zone 4 (cyan): Asian peninsula—Taiwan, Indonesia, Philipines, and Australia; zone 5 (purple): Pakistan, India, Bangladesh, Nepal, Egypt, and Sri Lanka; zone 6: Europe; zone 7 (brown): Madagascar; zone 8 (black): Africa; zone 9: North America; and zone 10 (orange): South America

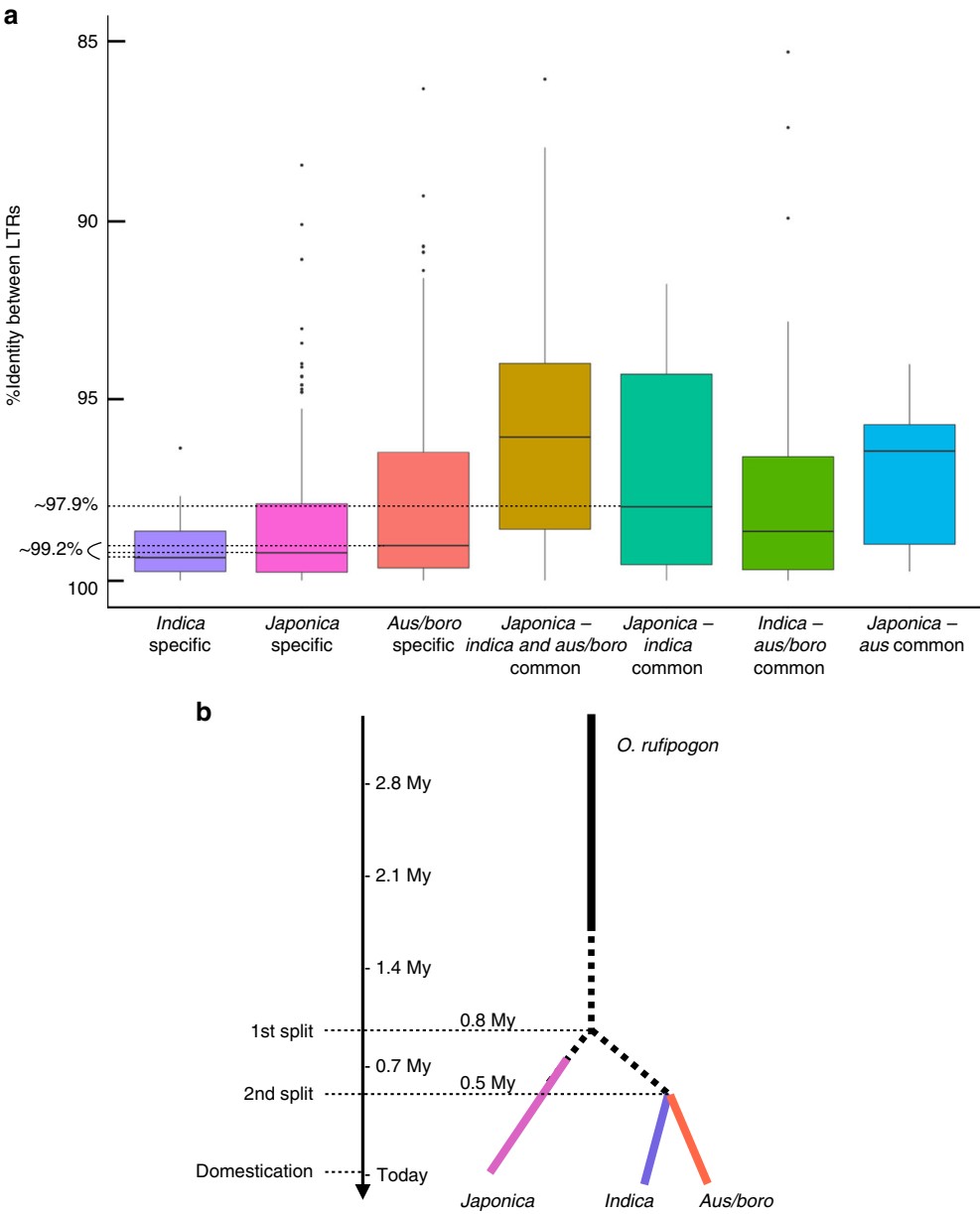

**Fig. 6** Origin of rice domestication. **a** The box plots represent the distribution of percent identity between LTRs of retrotransposons for insertions that are *Indica*-specific (*N* = 91), *Japonica*-specific (*N* = 266), *Aus/Boro*-specific (*N* = 216), common to the three groups (*N* = 138), common to *Indica/Japonica* (*N* = 32), common to *Indica/Aus* (*N* = 30), and common to *Japonica/Aus* (*N* = 12), respectively. The three horizontal dashed lines correspond to the mode of distribution of *Indica*-, *Japonica*-, and Aus-specific insertions. **b** Representation of the history of the three domestications of Asian rice

genomes of the wild progenitors of the three groups (Fig. 6b). The three values are significantly older than 10,000 years, i.e., the origin of rice cultivation at late Neolithic. The distribution median of *Indica–Japonica* common insertions is 97.9%, which translates into a date of 800,000 years ago. This confirms that the two gene pools of *Oryza rufipogon* from which both cultivated types originated were separated long before the origin of agriculture. This value also provides an estimate of the upper limit of the date of divergence between *Indica* and *Japonica* progenitors which, combined with the estimated date of the *Indica*- and *Japonica*-specific categories leads us to propose that the date of the divergence ranges from ~300,000 years ago to 800,000 years ago. The split of the *Indica/Aus* lineages appears to be more recent with a median at 98.6% (~540,000 years ago). Combined with the median age of the *Aus*-specific insertions (see above), our

results suggest that the split between the progenitors of *Indica* and *Aus* may have occurred between ~230,000 and ~540,000 years ago. However the median date of insertions that are common between *Japonica* and *Aus* (seventh group) appears to be older (96.4%, translating into a date of 1.4 Mya) than the median date of insertions that are common between *Japonica* and *Indica* (estimated above at 800,000 years ago). The much smaller sample of *Aus* varieties in the 1067 traditional cultivar's dataset (84 accessions) may explain this discrepency. Alternatively, it could originate from demographic history of the populations of the rice progenitor *O. rufipogon*. Further analyses may help clarifying this last point.

Our results, synthesized in Fig. 6b strongly suggest that *Indica*, *Japonica*, and *Aus/Boro* originated from three distinct domestication events. Unfortunately, the density of TIPs of all seven

categories was not sufficient to identify traces of introgression in the vicinity of domestication loci and therefore cannot solve the paradox of the presence of a single allele for these loci[34–36].

## Discussion

Several softwares designed for TIPs detection are currently available[26,27]. However, all require as a first step to map all sequencing reads onto a reference genome sequence, using Burrows–Wheeler-based algorithms. This mapping step is computationally intensive and thus the computation time necessary to analyze a large dataset (i.e., >1000 genomes) is too long. The availability of large datasets is now only a reality for a few plant species such as rice and *A. thaliana*, but this will certainly change in the near future. In this report, we show that TRACKPOSON, the new software we developed for identifying TIPs, is efficient for the analysis of thousands of genomes. It produces few false positives, as evidenced by wet-lab validation (Supplementary Figure 1 and Supplementary Tables 1 and 2). False negatives remain an issue with low genome coverage data; however, the two-pass procedure (see Results and Methods sections) can considerably limit the risk of missing TIPs, as confirmed by the sequencing of one accession with long-read technology (Nanopore). However, unlike the softwares cited above, TRACKPOSON is not designed for the exhaustive characterization of all structural variations caused by transposition, but rather for the complete and efficient characterization of the transpositional activity of known families among thousands genomes.

The 53,262 TIPs we identified in the 3000 rice genomes for 32 retrotransposon families were found at low frequency among rice accessions, similarly to what was observed for *A. thaliana*[45,50]. Most insertions are private or shared by two accessions, which suggests that they occurred after rice domestication and therefore in agro. This implies that retrotransposition-driven genomic diversification is ongoing in rice fields. We looked for genetic factors—the impediment of which may be causative of such activation, but found no evidence of mutations in genes of known epigenetic pathways involved in transposition control. Instead, the majority of peaks were found where a TIP of the same TE family was identified. From this, we propose that transposition in agro may require both the presence of an active TE copy in the genome and an external stimulus that may modify the epigenetic status of such copy, although the nature of such stimulus in rice is unknown. In particular, whether biotic and/or abiotic stresses are involved like in the case of other species remains to be elucidated[47].

We used retrotransposon insertions as paleogenomic tools to investigate the origin of rice domestication and clearly showed that *Indica*, *Japonica*, and *Aus/Boro* genomes diverged significantly earlier than rice domestication during late neolithic ca. 10,000 years ago. This finding contradicts the single origin hypothesis based on the identification of a single "domesticated" allele in most domestication loci. Although we cannot resolve this paradox, the most parsimonious interpretation of existing data is that at least three domestications occurred from three distinct gene pools of the wild rice progenitor *O. rufipogon*. The distribution of *O. rufipogon* in Asia today does not contradict this hypothesis because the species is now found all across South Asia, South-East Asia, and East Asia. Moreover, it is possible to identify populations that are closely related to each of the two cultivated types *Japonica* and *Indica*[38]. The presence of only one domesticated allele at a key domestication locus must then result from the introgression of one domesticated form by the other, thus explaining the loss of one of the two original alleles[37]. In support of this hypothesis, the current distribution of all varietal groups in ssAsia shows that they have been extensively disseminated

throughout the continent for a long time (Fig. 5b). Alternatively, a single allele may have been present in the wild progenitor and selected several times independently to give rise to *Japonica* and *Indica* rice[38].

The complete TIP data reported here is based on the analysis of 32 retrotransposon families. We chose a sample of families that is representative of the diversity found in the rice genome[16]. Other TE types, e.g., Miniature Inverted Transposable Element (MITEs), helitrons, and transposons should also be investigated in order to complete the characterization of the transpositional landscape of rice. The small size of many TE families of MITEs and Short Interspersed Element (SINEs) may however require to modify our detection method since the initial mapping step may be hindered by the small size of the element.

The advent of large genomic datasets opens new perspectives in the exploitation of genetic resources for most crops. For rice, the 3000 genomes are now being routinely used for gene discovery. However, most studies rely on the use of SNP data, while large InDels and in particular those caused by the activity of TEs remain mostly unexploited. In this study, we show that TEs are at the origin of a large extent of structural variations found genome-wide and arise at a rapid rate. Hence, these mobile elements contribute significantly to the genomic diversity of rice. We also show that most of this diversity originated after domestication, therefore in rice fields and wherever rice is grown, further suggesting that the whole rice gene pool may be a reservoir of structural variants. Whether such TE-driven genomic changes are phenotypically relevant and could therefore have been selected for as a source of adaptation for rice remains to be demonstrated, although evidence that TIPs are causal to phenotypes of agronomic interest have been reported for other crops. This dataset will allow others to explore the role of TIPs in rice diversification and breeding.

## Methods

**TRACKPOSON pipeline**. The TRACKPOSON pipeline is freely available at http://gamay.univ-perp.fr/~Panaudlab/TRACKPOSON.tar.gz. It is designed to detect TIPs in large datasets (>1000 genomes), using an existing TE database. The first step consists in mapping paired reads of genomic data in fastq format onto indexed consensus sequence of each TE family. Bowtie2 (v. 2.2.0) in very-sensitive mode[51] is used in this first step. The Sam file thus obtained is parsed using the flag value as a criterion as follows: only the read pairs for which one read mapped against the TE were kept. The next step consists of mapping the unmapped paired read onto the rice reference genome sequence (Nipponbare rice genome IRGSP1.0) using blastn (v. 2.2.31+)[52] with an *e* value threshold of $1e−20$. Only unique blast hits were considered and converted into bed format. In parallel, the Nipponbare genome (IRGSP1.0) was split into 10 kb windows by using bedtools makewindows (v. 2.25.0). Each TE insertion was thus assigned to a 10-kb window. Because a minimum of five amplicons spanning the insertion is required as an initial detection threshold, once a TIP had been detected in at least one accession, then the third step of the procedure consists in a new scan of all remaining accessions with a lower threshold (2 amplicons). Finally, a full matrix of presence/absence of TE insertions was created by using a home-made R (v. 3.3.1) script, with the 3000 rice genomes in columns and all the TIPs (for the TE family studied) in lines. These data are freely available at http://gamay.univ-perp.fr/~Panaudlab/TRACKPOSON_Results.tar.gz.

The wall time for running the software on the 3000 genomes ranged from a few hours to several days on a cluster of 88 cores, depending on the level of repetition of the family, confirming that this new strategy considerably improved the speed of detection.

**Mappability**. As mentioned in the paragraph above, only reads mapping unequivocally at a unique position onto the reference genome sequence were kept for positioning the TIPs. This obviously reduced the mappable genome to only unique sequences, which are estimated to be ~60% of the rice genome[15]. Each 10 kb window was sliced into 100 bp sequences—the repeat level of which was estimated using blastn against the reference genome with the same parameters as used in the mapping step of TRACKPOSON. Results were concatenated over each window to produce a percentage of mappability. These data were plotted onto the genome (Fig. 2).

**Validation TIPs with Nanopore sequencing**. We resequenced one *Indica* rice variety—i.e., IRIS-313–11419 using the Nanopore long-read technology. The

library was prepared using the 1D kit according to the manufacturer's instructions. One R9.4 flowcell was used. After basecalling of long reads with Albacore Oxford Nanopore software (to convert fast5 files in fasta format), a blast database was created. We performed a blastn of the TE families against this Nanopore database, with $e$ value $1e-50$ and penalities for open and extend gap equal to 0.

Only the reads with a High Scoring Pair (HSPs). corresponding to maximum of 80% of their length were kept. With this filtering, we eliminated the reads corresponding only to a TE. All the reads were validated by hand by dotter and NCBI blast against the rice reference genome IRGSP1.0. For each read thus selected, 300 bp of sequences flanking the insertion were used as query for blastn search against the IRGSP 1.0 rice pseudomolecules with $1e-50$ $e$ value threshold. This allowed unambiguous mapping of the TE insertions.

**Distance between TE insertion and gene estimation**. Estimation was performed with bedtools (v. 2.25) between the Nipponbare gtf annotation file (locus_IRSP1–0_predicted.gtf) and the output of TRACKPOSON pipeline (i.e., the TIPs localization).

**GWAS analysis**. For each of the 32 TE families, the number of TE insertions was determined for each variety by summing the number of TE copies identified. Thus, TE copy number was treated as quantitative phenotype from TE mobilization along the varieties. GWAS for TE copy number was carried out with a 404 K coreSNP dataset (160 K after minor allele frequency>5% and missing data<20%) downloaded from the Rice SNP-Seek Database[46] and using a linear-mixed model in EMMAX[53]. This procedure takes the underlying population structure into account by including a kinship matrix as a random effect. After checking the values of FDR (performed with Q-value R package), a stringent threshold of $-\log10 P = 6$ was set to declare a significant association. The GWAS interval was determined by taking into account the significant SNPs at each locus ($-\log10 P \geq 6$).

For characterization of the GWAS intervals, the TE annotation file and the Locus file containing all gene models were downloaded from the MSU Rice Genome Annotation Project v 7. All genomic annotations overlapping with these intervals were considered as putative causal genes. Regarding overlaps with TE, the GWAS intervals were compared to annotated TEs in the reference genome or the non-reference TE insertions identified by TRACKPOSON. The TE insertion that overlaps with GWAS intervals of the same family should be causal factors.

**Discriminent analysis**. DAPC was performed with the adegenet package (dapc function) in R (v. 3.3.1).

**Genomic paleontology**. Complete retrotransposon insertions were characterized in three well assembled rice genomes (*Japonica* Nipponbare, *Indica* IR8, and *Aus/Boro* N22) for nine of the most repeated families (i.e., *Dagul*, *Dasheng*, *Rire2*, *Rire3*, *Hopi*, *Houba*, *RN215*, *Osr37*, and *Poprice*). For this, a home-made script perl (available upon demand) was used. Briefly, for each family, three sequences are first generated, i.e., the full element, the LTR sequence, and the portion of the internal sequence that corresponds to the RT domain (except for the LARD dasheng for which a non-complex portion of the internal sequence was used). The first step consists in looking at all the paralogs of the element using the RT domain as a query in a blast search against the reference genome, each hit being extended both upstream and downstream, and the resulting genomic sequence checked for the presence of both LTRs (using blastn with the LTR of the consensus sequence against the genomic region obtained above). After trimming the sequence, the identity between the two LTRs was estimated by splitting the element into two equal sequences and blasting one half against the other (only the LTRs produce alignments, from which a percentage of identity was obtained from the largest HSP).

Once all complete elements were identified in both genomes, their orthologous relationship was secured as follows: 300 bp of genomic sequence upstream of the element was extracted from the first reference genome sequence and used as query in a blastn search against the other genome. Only sequences producing a single hit with at least 90% of the query length were considered orthologous and therefore kept for further analyses. The presence of the element at an orthologous position in the other genome was checked by first extracting 15 kb of the sequence downstream of the 300 bp and blasting the resulting sequence against the consensus sequence of the TE family. Therefore, each insertion could be classified as *Japonica*-specific (present in Nipponbare and absent from IR8 and N22), *Indica*-specific (present in IR8 but absent from Nipponbare and N22), *Aus*-specific (present in N22 but absent from Nipponbare and IR8), common to all (present in the three genomes), or common to two of the three genomes (three classes). This classification was finally validated as follows: *Indica*-, *Japonica*-, and *Aus*-specific insertions were retained only if present in at least 60% of varieties from the corresponding group, and the common insertions should be present in at least 80% of the varieties of the groups.

**Reporting summary**. Further information on experimental design is available in the Nature Research Reporting Summary linked to this article.

## Data availability

The Nanopore sequencing data is available on NCBI under the BioProject ID PRJNA507708. The previously published 3000 rice genome raw sequencing data are available from GigaScience Database (https://doi.org/10.5524/200001)[23]. A reporting summary for this Article is available as a Supplementary Information file. The source data underlying Fig. 2, Fig. 3a–c, Fig. 5, Supplementary Figure 1, Supplementary Figure 2, and Supplementary Table 3 are provided as a Source Data file.

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

## Acknowledgements

This work was supported by a CNRS/Région Languedoc Roussillon research grant and by the University of Perpignan via domitia. This work was publicly funded through a CNRS/Région Languedoc Roussillon research grant, by the University of Perpignan via domitia and ANR (the French National Research Agency) under the "Investissements d'avenir" programme with the reference ANR-10-LABX-001–01 Labex Agro and coordinated by Agropolis Fondation. The work was also supported by an Academia Sinica Thematic Project AS-TP-107-L02 in Taiwan. The authors thank Marie Mirouze, Scott A. Jackson, Josep Casacuberta, and Joris Bertrand for their useful comments on the manuscript.

## Author contributions

M.-C.C. designed and built the TRACKPOSON software, performed TIP detection, contributed to the other analyses, and wrote the manuscript. E.M. and R.A. conducted GWAS. F.-J.W. and H.-P.W. implemented the software in the computer facility at Academia Sinica (Taipei, Taiwan). E.L. and E.D. performed the wet-lab experiments and analyses for the validations of TIPs. C.L. performed the Nanopore sequencing. Y.-I.H. supervised the work at Academia Sinica. O.P. supervised the project and performed the genomic paleontology analyses.

## Additional information

**Competing interests:** The authors declare no competing interests.

