## [Peer Review File · Nature Communications]

Reviewers' comments:

Reviewer #1 (Remarks to the Author):

Carpentier et al. describe the genome-wide analysis of transposable element insertion polymorphisms (TIPs) of 31 L1-retrotransposon families in 3000 rice genomes (including 1067 traditional varieties). As no gene involved in transposon control was found to be significant in a GWAS using 'number of copies' as phenotype, the authors conclude that external stimuli are responsible for transposon activation. Several transposon families are dominated by singleton TIPs indicating on going transposon activation. Finally the authors dated insertions for 9 TE families with high TIP count to estimate the split of Japonica and Indica (>100k years) – showing that the split was before rice cultivation. This TE landscape analysis in a large cohort is highly interesting, well written and the statistical methods seem to be appropriate. The results are of general interest.

In addition the authors report on their new method for transposon insertion detection in large cohorts. This part of the manuscript is a bit weaker and quite lengthy considering the limited novelty. The mapping of reads against a TE-sequence library instead of the whole genome is presented as major difference to other tools, although this can easily be accomplished by existing tools by simply swapping the reference file for a TE file (and has also been done before by e.g. VariationHunter, hence not novel). Moreover the new method TRACKPOSON is not benchmarked against existing methods, and therefore it is unclear if it underperforms or not. In my opinion this part of the manuscript could either be shortened to make it mainly a data analysis paper, or a proper benchmark with at least 1 other tool (on a subset of samples to reduce computation time) should be added. The latter option would be better as it would raise the confidence in the presented results.

Here are some criticisms and questions, mostly focusing on the method (as the statistical analysis part of the manuscript looks sound to me as it is):

1. Presenting a new tool (as major part of the main manuscript) without a benchmark against existing methods/tools is not state-of-the art. It also lowers the confidence in the subsequent analysis that is based on the identified TIPs.
2. The authors state that most TIPs are singletons: but these could also be frequent false positives. A random subset of these singleton TIPs should be validated by Sanger sequencing to estimate the false positive rate for singletons (which could be higher, as is well known for SNP detection). Or if there is long-read data for at least one sample this could be used for validation as well. Alternatively other tools could be used to show that these tools also identify the singletons (although this is not as convincing as a Sanger/long-read overlapping the breakpoints)
3. Are there no BAM files available for the 3000 rice samples or at least for a subset? This would substantially reduce the computation time needed to benchmark against existing TE tools. Especially it would also help to prove that the TE-library-mapping method is not substantially worse than reference-mapping methods.
4. At the end of the discussion the authors state that there are not just 31, but around 300 LTR-retrotransposon families. But the authors do not explain why they did not analyze all 300? Could this be elaborated on? Are these TE families less frequent, or less important? Also, why is there a problem with shorter TEs? Modern mappers like bwa-mem can do (soft-)clipped mapping if the read is longer than the TE or overlaps the breakpoint. Please discuss.
7. Why is blastn used for mapping of TE-paired reads to the reference? Could the nucleotide divergence be too high for bwa-mem or bowtie2 to correctly map the read? After all blastn is much slower. Moreover, is the 10kb resolution good enough to make sure that the same TE insertion site

was detected in two varieties (i.e. would breakpoint resolution be better)?

Reviewer #2 (Remarks to the Author):

In this manuscript, Carpentier et al. take advantage of a large dataset of publically available genome sequences (3000 in total) of wild and cultivated rice to determine the landscape of insertion polymorphisms (TIPs) for 31 of the approximately 300 retrotransposon families identified so far in the genome of this species. Using a custom-designed pipeline, TRACKPOSON, which enables high-throughput analysis of non-reference transposable element insertions in large genomes, they identified more than 50,000 TIPs, most of which low frequency variants, suggesting that these 31 retrotransposon families are active in cultivated rice. Moreover, the authors show that the size of each of these 31 families varies between varieties and that this size variation is associated, at least for the 12 smallest families species-wide, with cis DNA sequence polymorphisms. Finally, the authors use the level of sequence identity between LTRs of full-length LTR-containing retroelement insertions to estimate the age of these insertions and found as expected that older and younger insertions tend to be high and low frequency variants, respectively. Furthermore, age estimates are congruent with a diphyletic origin of domesticated rice.

Although this is an interesting manuscript, many of the results are over interpreted and key analyses are missing, as detailed below.

1) The performance of TRACKPOSON needs to be fully evaluated. The comparison made by the authors with their previous pipeline (PEM-based software) suggests a low percentage of false negatives but no information is provided as to the percentage of false positives, which is required to calculate the FDR. Furthermore, this comparison is unsatisfactory because it does not involve a non-reference genome sequence and because of the 3000 non-reference genomes that are the basis of the current study have much reduced sequence coverage (11.6X on average vs 30X for the first comparison). The authors should first evaluate the sensitivity and specificity of TRACKPOSON by subsampling the 30X WGS data set for the reference genome (i.e. 33% of the raw data). Second and most importantly, the authors should take advantage of the three high quality PacBio-based genome assemblies available (Schatz et al 2014, Genome Biol) to assess directly the false positive and negative rate of TRACKPOSON. Finally, it is not clear if TRACKPOSON offers a real improvement over other pipelines designed to identify TIPs, since most research centers have access to large computing clusters (>1000 CPUs), which considerably reduce wall time.

2) The authors provide only a partial view of the landscape of retrotransposon insertion polymorphisms (TIPs). This should be made clear at the outset (in the title, abstract and the Introduction). The authors should explain why they limited their analysis to only 31 of the approx. 300 retrotransposon families annotated in the rice genome and did not consider analyzing also DNA transposon families. Furthermore, as it stands, the manuscript does not present the "retrotranspositional landscape" for the 31 families analyzed, as old and recent TIPs are considered together (Figure 2), thus with no attention paid to the role of selection (natural as well as artificial) and demography in shaping the TIP landscape. In this context, it is not clear whether or not one should be surprised that most TIPs are low frequency variants. Indeed, it is well established in plants and animals that TIPs are typically low frequency variants.

3) To determine the retrotranspositional landscape, the authors should focus on private and rare TIPs, most of which should reflect recent retrotransposition activity. The authors should analyze their distribution along the genome (Figure 2) and in relation to genes (Table 1) and contrast the results obtained with those obtained for frequent TIPs and fixed or nearly fixed insertions (frequent TIPs should indeed be distinguished from fixed or nearly fixed insertions). These additional analyses will in turn help identify potential demographic effects (or potential positive selection), in

the form of a statistically significant enrichment of frequent TIPs as close to genes as private or rare TIPs.

4) The authors performed GWAS to identify the genetic determinants of variable retrotransposition activity for the 31 families analyzed. Robust cis associations (cis SNPs) were identified for the 12 less repeated families, thus providing strong evidence that genetic factors do modulate retrotransposition activity in rice, contrary to what the authors state. Many of these cis SNPs should map within TIPs (as found in *Arabidopsis thaliana*, Quadrana et al, eLife 2016) and the authors should examine such variants for potential causal SNPs. However, because GWAS is underpowered to detect associations with low frequency variants, some GWAS intervals spanning TIPs can be the result of sequence variation among haplotypes lacking the TIPs. The authors should therefore determine if the presence/absence and frequency of potentially causal TIPs are compatible with the haplotypes carrying the associated SNPs.

5) The lack of association with SNPs located within or close to genes involved in epigenetic silencing of TEs is not surprising, given that mutations in these genes are likely strongly counter selected. Finally, the absence of trans associations cannot and should not be taken as evidence that TE mobilization is determined by environmental factors. The authors should therefore correct the manuscript (including the abstract) accordingly; as no evidence at all is presented that support their claim of environmental influences.

6) The use of retrotransposon insertions as paleogenomic tools to investigate the origin of rice domestication is interesting. However, determining the age of TE insertions based on LTR sequence identity and nuclear mutation rate is a very rough approximation as retrotransposition is error-prone (Drake et al 1998, Genetics). This should be taken into account and/or discussed. Also, the authors should discuss their results in light of the large body of literature supporting a multiple origin of cultivated rice and a single domestication event with subsequent introgressions of key domestication alleles (Huang et al Nature 2012, Gross et al PNAS 2014, Huang et al Nature Plants 2015, Choi et al MBE 2017, none of which are cited here).

7) The authors should provide detailed information (including URL of public repositories) of the published datasets used in this work. Additionally, a file containing the genomic coordinates as well as presence/absence of TIPs across the rice varieties needs to be presented.

Other points

8) Table 1 contains 32 TE families, not 31.

9) In several parts of the manuscript the authors state that they analyzed LTR-retroelements, however they also include the analysis of KARMA, which is a non-LTR retrotransposon. Please correct and explain why you specifically picked this non-LTR retrotransposon.

10) Many of the violin plots in Figure 6 show data above 100% identity. Please revise.

11) Ref. 14 in the "GWAS analysis" section of Materials and Methods is not the right one.

12) What are the GWAS intervals (page 14?).

13) The study of Stuart et al 2016 eLife should be also cited when discussing TIP frequency in *A. thaliana* page 10).

14) The text needs language editing (e.g. evidence is always singular; insertions present in single accessions are private, not shared; it is not clear what "this process" refers to in the second paragraph of the introduction, etc.)

Reviewer #3 (Remarks to the Author):

Review for "Retrotranspositional landscape of Asian rice (*Oryza sativa*) revealed by 3,000 genomes".

This is a first genome wide scale analysis of retrotransposon polymorphism using a very large rice population. The authors developed novel software for this specific task. Using this tool, the author determined the frequency of polymorphic insertions, the possible factors that may influence the transposition, and they deduced the genetic distance between japonica and indica populations. The knowledge gained from this approach cannot be obtained in other ways. Particularly, I think it is interesting to observe that in most cases the GWAS peak for copy number of a transposon is overlapping with one or more insertions of the particular element. As a consequence, I think the most of the results from this study is very inspiring. However, I do have problems with the interpretation of some of the data.

1. One domestication or two domestication events (or multiple events)

I guess this is the most controversial topic in the rice community and I need to clarify that I am not a fan of either sides. The authors of this study clearly show that the gene pools for indica and japonica separated long time before domestication and they really deserve some credit for this. However, I have to argue this does not really provide direct evidence for two distinct domestication events. According to the one domestication hypothesis, the initial domestication event led to the formation of japonica, and then japonica crossed with local wild rice to form indica cultivars. Certainly the feasibility of the proposed path is questionable (but not impossible). Regardless, if it indeed occurred, we would probably see the same results as we see here. This is because, after two backcrosses, the majority of donor-specific TIPs would be lost from the genome – especially if we consider most of them are probably not selected for. Certainly the most parsimony explanation for the observation made in this study is that there were two (or maybe more) domestication events. However, when it was mixed with artificial selection, human migration, natural and artificial introgression, parsimony may not explain everything. So "confirm that rice originated from two distinct domestication events." is really way too strong. So my suggestion is that the author should turn the tune down on this issue.

2. p.6 "However, the mapping of all TIPs on the 12 rice pseudomolecules shows no insertion bias at the chromosome level (Figure 2)"

I looked at Figure 2. It seems to me for most chromosomes, the gypsy TIPs demonstrate a bias toward centromeric and pericentromeric regions as well as short arms of chr04 and chr10. These regions are known to be rich on gypsy elements. If the authors want to keep this statement, there should be a statistic test of comparison between chromosomal arms and pericentromeric regions.

3. p.6 bottom "Furthermore, we did not find any difference between Gypsy and Copia elements in respect to the distance of insertion from genes (Table 1).

First of all, it is unclear to me whether the authors refer to all elements or just TIPs, so please clarify. Second, I am very much surprised that there is no difference between gypsy and copia. There are multiple studies indicating that copia elements are closer to genes than gypsy elements. I did not find any details about how they calculate the distance. My suggestion is, the authors should double check everything. Particularly, it is important to exclude transposon genes from the current gene set. If they still get the same results, they should discuss why their result is different from previous results.

4. p.8 middle "ar at least which may be active in agro since its presence is" should be "or at least ...".

Reviewer #4 (Remarks to the Author):

This paper provides a retrotransposon-focused analysis of a population-scale genomics dataset of cultivated rice (*Oryza sativa* L.) recently published by the 3,000 Rice Genome Project. The impressive size of the dataset is unprecedented in crop research, which poses specific computational challenges for genomic analyses, but also promises new discoveries and insights into genomic organisation, evolution, population history etc. Transposable elements (TEs) comprise large fractions of crop genomes which are no longer considered irrelevant in respect to genome organisation and complex regulation of gene expression, and consequently, phenotypic variation. It is therefore of high importance to characterise TE profiles and the dynamics of TE evolution in crop species.

Through clearly defined objectives, appropriate methodology and good organisation, this paper presents the following new findings and technical advances:

(I) new software TRACKPOSON designed for detection of TE insertion polymorphisms (TIPs) in large datasets

(II) most of TIPs were found at very low frequencies, which is interpreted as evidence of very recent activity (following domestication)

(III) GWAS did not identify clear associations of nucleotide variants and the TE copy numbers, which was interpreted to suggest TE proliferation as a response to environmental stimuli

(IV) application of the molecular clock principle resulted in dating the split between indica and japonica gene pools long before the beginnings of agriculture, which is interpreted as evidence of two separate domestication processes

In respect to (I), I think that TRACKPOSON offers a clever way how to speed-up the TIPs detection, thereby allowing such analysis in population-scale whole-genome datasets. I don't see why mapping all data onto the genome reference (the first step of alternative methods) could have any advantages for the TIPs detection, therefore I think that removing or reducing this step is safe and should not affect sensitivity or specificity of the detection.

Question: I don't see how TRACKPOSON (or similar methods) could distinguish two insertions of the same class TE in two individuals at nearby locations. E.g., two Hopi retrotransposons are inserted upstream of a gene XY, each insertion occurring in different individuals but very similar (non-identical) location. Are such insertions correctly resolved as unique, or incorrectly as a single event?

(II) The low frequencies of most TIPs are really intriguing and I am inclined to accept authors' interpretation that this implies their origin in agro. However, in my understanding, TIPs do not only result from insertions of new TE copies, but also from removal of the old ones. The removal is recombination dependent and usually results in a single LTR remnant.

Can we be sure that the low frequencies of TIPs reported here are not due to a very dynamic TE-removal process? Many older TEs can be completely removed, hence undetected, while others are removed incompletely and persist in the population, albeit in low frequencies. Does the detection of TIPs associated with single LTRs and complete TEs have different sensitivities? Can you distinguish TIPs associated with full TEs from those associated with single LTRs?

(III) I think this is technically good. However, I find the wording of the conclusions a bit confusing. The authors write: "we also show that the activation of transposition in rice is probably triggered by an external stimulus, rather than by the impediment of a silencing pathway". In my understanding, the authors are trying to differentiate between genetic mutations in genes involved in the epigenetic modification pathways (methyltransferases or RNA-directed DNA methylation pathway) – and external stimuli, i.e. biotic and abiotic stresses. However, it appears that the environmental stresses do not cause demethylation directly – but rather indirectly by modulating the DNA methylation pathways (Downen et al. 2012; Luna et al. 2012; Yu et al. 2013). This process may be governed by salicylic acid-mediated stress response. Hence, the "external stimulus",

“impediment of a silencing pathway” and demethylation are all interconnected and I find it a bit confusing to put them in contrast. I suggest to rephrase.

(IV) This part of the manuscript does not bring much novelty, but can be significantly improved by additional analysis (see below). First, the conclusion that the genomes of indica and japonica are separated by several hundred thousand years is old (Ma & Bennetzen 2004; Vitte et al. 2004 – the same senior author as in this manuscript) and generally accepted. However, it is becoming clear that *O. sativa* includes a third group – aus – and it is not yet widely accepted that aus and indica represent two gene pools with separate origins. The 3,000 RGP dataset contains >200 aus accessions. Expanding the analyses to include this group could help resolve the issue of aus origin. I would be very excited to see the dating of the indica-aus split, and this could have very high impact.

Secondly, the debate of indica-japonica domestication has moved away from the genome-wide comparisons. The observation that the “genomic backgrounds” (the backgrounds of the domestication genes) in indica and japonica are different has been supported by solid evidence. Currently – as the authors rightly noted – the question of domestication revolves around the genealogical origin of the domestication genes. Proponents of the introgression hypothesis claim that a few genes critical for the domestication phenotype were transferred from japonica to indica by introgression, and hence the domestication phenotype originated in a single domestication event and was later transferred into proto-domesticated or wild indica. Although I do not agree with that model and see it as an ad hoc hypothesis to save the single domestication model from being falsified, it is still a plausible scenario. Importantly, if it is correct, then the observation of indica-japonica differences at the genome-wide level does not prove independent domestications. Instead, focusing on the regions surrounding the domestication genes is essential to resolve this problem; however, the authors say that the TIPs do not have sufficient densities for such comparisons. Taking all these points into account, the claim “rice originated from two distinct domestication events” is unsubstantiated and should be changed to something like “rice originated from at least two distinct gene pools”.

In relation to this, the authors also say “These two gene pools must have split several hundreds of thousand years ago, possibly due to the rise of the Himalayas.” I find this idea geographically confused. The Himalayas split the Asian continent into a southern part (the Indian subcontinent) and the northern part. However, wild rice does not grow north of the Himalayas, and the indica-japonica gene pool split is rather longitudinal – with the progenitor of japonica found in southern and eastern China, and the progenitors of aus and indica in the Indian subcontinent and Indochina (Civan et al. 2015; Civan and Brown 2018).

The authors also say “Alternatively, one could also argue that the domesticated alleles have been present in the populations of the wild progenitor of rice long before the split of the two gene pools that gave rise to the domesticated forms and then selected for two times independently.” This was first suggested in Civan and Brown 2017 and later confirmed in Civan and Brown 2018; and it is also consistent with the results of Wang et al. 2018. I think these works should be referenced.

Other questions:

The authors say that they analysed 1067 traditional varieties included in the 3,000 genomes. However, the identities of the 1067 traditional varieties are not given, and I cannot find such characterisation in the online resources of the 3k RGP dataset. Can you provide a list?

Peter Civan, 14. 05. 2018, Manchester, UK

References:

- Downen et al. 2012. Proc Natl Acad Sci U S A. 109:E2183–E2191.
- Luna et al. 2012. Plant Physiol 158:844–853.
- Yu et al. 2013. Proc Natl Acad Sci U S A. 110:2389–2394.

Ma & Bennetzen 2004. *Proc Natl Acad Sci U S A.* 101:12404–10.
Vitte et al. 2004. *Mol Gen Genom.* 272:504–11.
Civan et al. 2015. *Nat plants* 1:15164.
Civan & Brown 2017. *Genet Resour Crop Evol.* 64:1125–1132.
Civan & Brown 2018. *BMC Evol Biol.* 18:57.
Wang et al. 2018. *Nature.* 557, pages43–49

REPLY to REVIEWERS :

Reviewer #1 (Remarks to the Author):

1. Presenting a new tool (as major part of the main manuscript) without a benchmark against existing methods/tools is not state-of-the art. It also lowers the confidence in the subsequent analysis that is based on the identified TIPs.

We have benchmarked TRACKPOSON against the recently released TE detection software JITTERBUG (Henaff et al., BMC genomics 2015) and emphasized throughout the manuscript that the main objective of TRACKPOSON is the fast detection of known TE families.

2. The authors state that most TIPs are singletons: but these could also be frequent false positives. A random subset of these singleton TIPs should be validated by Sanger sequencing to estimate the false positive rate for singletons (which could be higher, as is well known for SNP detection). Or if there is long-read data for at least one sample this could be used for validation as well. Alternatively other tools could be used to show that these tools also identify the singletons (although this is not as convincing as a Sanger/long-read overlapping the breakpoints)

We have sequenced one variety that belongs to the 3k sample using long reads (Nanopore) to perform a wetlab validation of the detection. We now have a better estimation of both specificity and sensitivity of TRACKPOSON.

3. Are there no BAM files available for the 3000 rice samples or at least for a subset? This would substantially reduce the computation time needed to benchmark against existing TE tools. Especially it would also help to prove that the TE-library-mapping method is not substantially worse than reference-mapping methods.

To our knowledge and at the time the analysis was done, no such resource was available. In addition, as mentioned in the manuscript, we anticipate that genomic resources for large samples will soon become available for many plants and animals and we thus endeavoured to develop a simple tool that can quickly assess the transpositional activity of a set of known TE families at population level. The generation and storage of BAM files for thousands of individual may not be as easily accessible as the raw sequence data.

4. At the end of the discussion the authors state that there are not just 31, but around 300 LTR-retrotransposon families. But the authors do not explain why they did not analyze all 300? Could this be elaborated on? Are these TE families less frequent, or less important? Also, why is there a problem with shorter TEs? Modern mappers like bwa-mem can do (soft-)clipped mapping if the read is longer than the TE or overlaps the breakpoint. Please discuss.

We modified the text accordingly, stating that we chose a sample of LTR retrotransposons that are representative of the 300 families. Therefore we do not claim that we fully characterized the retrotranspositional landscape of the rice genome, but instead unraveled some features of the retrotranspositional landscape. As for shorter TEs, we in fact started to work on some MITEs but encountered some mapping issues. The main problem is that the reads obtained in the frame of the 3k genome project are short (<100 bp). We are now working on this issue.

7. Why is blastn used for mapping of TE-paired reads to the reference? Could the nucleotide divergence be too high for bwa-mem or bowtie2 to correctly map the read? After all blastn is much slower. Moreover, is the 10kb resolution good enough to make sure that the same TE insertion site was detected in two varieties (i.e. would breakpoint resolution be better)?

BLASTn is used because indeed it better buffers sequence variation in the flanking sequence of the insertion. The use of bowtie2 indeed lead to higher false negative rate.

Reviewer #2 (Remarks to the Author):

1) *The performance of TRACKPOSON needs to be fully evaluated. The comparison made by the authors with their previous pipeline (PEM-based software) suggests a low percentage of false negatives but no information is provided as to the percentage of false positives, which is required to calculate the FDR. Furthermore, this comparison is unsatisfactory because it does not involve a non-reference genome sequence and because of the 3000 non-reference genomes that are the basis of the current study have much reduced sequence coverage (11.6X on average vs 30X for the first comparison). The authors should first evaluate the sensitivity and specificity of TRACKPOSON by subsampling the 30X WGS data set for the reference genome (i.e. 33% of the raw data). Second and most importantly, the authors should take advantage of the three high quality PacBio-based genome assemblies available (Schatz et al 2014, Genome Biol) to assess directly the false positive and negative rate of TRACKPOSON.*

We have sequenced one of the non-reference genome from the 3k sample, using Nanopore long read technology. We chose an accession that was not well covered (ie 8X). We obtained an 11x coverage long reads sequences and could therefore manually check the rate of both false positives and false negatives (using previously published TE detection method for nanopore, Debladis et al., BMC genomics 2017). This is now included in the manuscript and surely makes the data of the 3k more robust.

Finally, it is not clear if TRACKPOSON offers a real improvement over other pipelines designed to identify TIPs, since most research centers have access to large computing clusters (>1000 CPUs), which considerably reduce wall time.

We anticipate that genomic resources for large samples will soon become available for many plants and animals and we therefore endeavoured to develop a simple tool that can quickly assess the transpositional activity of a set of known TE families at population level. Softwares that use PEM and SR methods will require the generation and storage of thousands of BAM files for each population genomics project. While there is no doubt that large computing clusters can cope with existing project such as the 1k arabidopsis or the 3k rice projects, the drop in sequencing cost leads to an exponential increase of sequence data for many plants and animals and I seriously doubt that existing clusters are properly sized for managing such amount of data with existing softwares.

2) *The authors provide only a partial view of the landscape of retrotransposon insertion polymorphisms (TIPs). This should be made clear at the outset (in the title, abstract and the Introduction). The authors should explain why they limited their analysis to only 31 of the approx. 300 retrotransposon families annotated in the rice genome and did not consider analyzing also DNA transposon families.*

We modified the text accordingly, stating that we chose a sample of LTR retrotransposons that are representative of the 300 families. Therefore we do not claim that we fully characterized the retrotranspositional landscape of the rice genome, but instead unraveled some features of the retrotranspositional landscape. As for shorter TEs, we in fact started to work on some MITEs but encountered some mapping issues. The main problem is that the reads obtained in the frame of the 3k genome project are short (<100 bp). We are now working on this issue.

Furthermore, as it stands, the manuscript does not present the “retrotranspositional landscape” for the 31 families analyzed, as old and recent TIPs are considered together (Figure 2), thus with no attention paid to the role of selection (natural as well as artificial) and demography in shaping the TIP landscape. In this context, it is not clear whether or not one should be surprised that most TIPs are low frequency variants. Indeed, it is well established in plants and animals that TIPs are typically low frequency variants.

3) *To determine the retrotranspositional landscape, the authors should focus on private and rare TIPs, most of which should reflect recent retrotransposition activity. The authors should analyze their distribution along the genome (Figure 2) and in relation to genes (Table 1) and contrast the results obtained with those obtained for frequent TIPs and fixed or nearly fixed insertions (frequent TIPs should indeed be distinguished from fixed or nearly fixed insertions). These additional analyses will in turn help identify potential demographic effects (or potential positive selection), in the form of a statistically significant enrichment of frequent TIPs as close to genes as private or rare TIPs.*

This is a very good point and I agree that this may be considered as a weakness of our manuscript that I (O. Panaud) am fully aware of. I may however argue that there are several issues that should be taken into account :

We recently showed that TE-driven genomic turn over is very fast in *Oryza* genus, with a half life of ~1My for most retrotransposons (Stein et al., Nature Genetics 2018). This is obviously for insertions that have not been counterselected and that can still be found at orthologous position in A-genome species of *Oryza*, i.e. cultivated rice and closely related wild species. We also suggested earlier that the main elimination force of TE insertions is indeed selection (Vitte et al., BMC genomics 2007). Therefore, one should consider both forces when analyzing TE dynamics at interspecific level. Some insertions are not rare in the 3k sample. These are probably neutral. This is expected in a 400 Mbp genome as most insertions will fall in intergenic space with no effect on gene expression, unlike in the case of *Arabidopsis thaliana*. Keeping in mind that domestication occurred (only) 10,000 years ago, one should expect that most of these neutral insertions should be intact (with very few substitution or deletions) and therefore easily detected using TRACKPOSON. As for low frequency insertions, I fully agree with reviewer 2 that the results should theoretically be interpreted in the light of both selection and demography. However, in the case of a domesticated species, demography is very difficult to cope with : as shown in figure 5B there has been a lot of dissemination, mixing and probably introgressions among rice populations over the last 10,000 years, as a result of complex human migrations in Asia. I must humbly admit that I have no idea how to take this into account analytically without more local sampling of traditional varieties. As it stands the 3k sample is a good representation of the overall diversity of Asian rice, but not very useful to fully understand micro-evolution at regional scale. Regarding selection, I am sure that reviewer 2 shares the same interpretation as me : the high rate of low frequency insertions shows that most insertions are efficiently eliminated from gene pools. (the alternative non parsimonious explanation being that transposition has started to occur within the last hundreds of years only). We chose to not address this point in the discussion because we feel that we do not have enough experimental evidences to claim this and surely, this could have been regarded as over-interpretation of data.

As for the retrotranspositional landscape issue, I don't fully agree with reviewer 2 that only rare and private alleles should be taken into account. A plant genome is mainly shaped by transposition :

- 1- with neutral insertions being retained for few 100,000 years (or more in some lineages like in gymnosperms). These may turn out to be at the origin of chromosomal rearrangements (eg through ectopic recombination) and should therefore be taken into account as a factor of genome differentiation.

- 2- with non-neutral insertions that are either quickly eliminated or retained through selection.

In my opinion, both are equally important and participate to shaping the genome.

4) *The authors performed GWAS to identify the genetic determinants of variable retrotransposition activity for the 31 families analyzed. Robust cis associations (cis SNPs) were identified for the 12 less repeated families, thus providing strong evidence that genetic factors do modulate retrotransposition activity in rice, contrary to what the authors state. Many of these cis SNPs should map within TIPs (as found in Arabidopsis thaliana, Quadrana et al, eLife 2016) and the authors should examine such variants for potential causal SNPs.*

How could cis-SNPs map within TIPs since these insertions are absent from the reference genome ? While I fully agree that the corresponding insertion may be divergent to some extent from the consensus copy of the family, I don't see how such SNP could be present in the 3k SNP dataset since it could not have been unambiguously mapped onto the reference genome (where it is not present). This is clearly the case for the *Tos17* insertion of chromosome 7, strongly associated with copy number but absent from the Nipponbare genome.

5) *The lack of association with SNPs located within or close to genes involved in epigenetic silencing of TEs is not surprising, given that mutations in these genes are likely strongly counter selected. Finally, the absence of trans associations cannot and should not be taken as evidence that TE mobilization is determined by environmental factors. The authors should therefore correct the manuscript (including the abstract) accordingly; as no evidence at all is presented that support their claim of environmental influences.*

We agree that we over-interpreted the data and have rewritten the discussion.

6) *The use of retrotransposon insertions as paleogenomic tools to investigate the origin of rice domestication is interesting. However, determining the age of TE insertions based on LTR sequence identity and nuclear mutation rate is a very rough approximation as retrotransposition is error-prone (Drake et al 1998, Genetics). This should be taken into account and/or discussed. Also, the authors should discuss their results in light of the large body of literature supporting a multiple origin of cultivated rice and a single domestication event with subsequent introgressions of key domestication alleles (Huang et al Nature 2012, Gross et al PNAS 2014, Huang et al Nature Plants 2015, Choi et al MBE 2017, none of which are cited here).*

The origin of rice domestication has been debated for years. For years, I have asked my colleagues in the rice community to explain the fact that indica- and japonica- specific ClassI TE insertions are disseminated throughout the 12 chromosomes of rice. That indeed would only be possible if introgressions would have literally replaced a genome by another one except for the domestication gene.... We have included the references + discussed the point raised by reviewer 2.

7) *The authors should provide detailed information (including URL of public repositories) of the published datasets used in this work. Additionally, a file containing the genomic coordinates as well as presence/absence of TIPs across the rice varieties needs to be presented.*

This is done.

Other points

8) *Table 1 contains 32 TE families, not 31.*

Done

9) *In several parts of the manuscript the authors state that they analyzed LTR-retroelements, however they also include the analysis of KARMA, which is a non-LTR retrotransposon. Please correct and explain why you specifically picked this non-LTR retrotransposon.*

This is corrected and we included the reference on Karma showing that it is transpositionally active.

10) *Many of the violin plots in Figure 6 show data above 100% identity. Please revise.*

Done

11) *Ref. 14 in the "GWAS analysis" section of Materials and Methods is not the right one.*

Done

12) *What are the GWAS intervals (page 14?).*

Done

13) *The study of Stuart et al 2016 eLife should be also cited when discussing TIP frequency in A. thaliana page 10).*

Done

14) The text needs language editing (e.g. evidence is always singular; insertions present in single accessions are private, not shared; it is not clear what “this process” refers to in the second paragraph of the introduction, etc.)

Done + final editing will be made according to the editor of the journal, if accepted for publication.

Reviewer #3 (Remarks to the Author):

1. One domestication or two domestication events (or multiple events)

I guess this is the most controversial topic in the rice community and I need to clarify that I am not a fan of either sides. The authors of this study clearly show that the gene pools for indica and japonica separated long time before domestication and they really deserve some credit for this. However, I have to argue this does not really provide direct evidence for two distinct domestication events. According to the one domestication hypothesis, the initial domestication event led to the formation of japonica, and then japonica crossed with local wild rice to form indica cultivars. Certainly the feasibility of the proposed path is questionable (but not impossible). Regardless, if it indeed occurred, we would probably see the same results as we see here. This is because, after two backcrosses, the majority of donor-specific TIPs would be lost from the genome – especially if we consider most of them are probably not selected for.

Certainly the most parsimony explanation for the observation made in this study is that there were two (or maybe more) domestication events. However, when it was mixed with artificial selection, human migration, natural and artificial introgression, parsimony may not explain everything. So “confirm that rice originated from two distinct domestication events.” is really way too strong. So my suggestion is that the author should turn the tune down on this issue.

We agree with reviewer 3 that We rephrased our discussion and conclusion.

2. p.6 “However, the mapping of all TIPs on the 12 rice pseudomolecules shows no insertion bias at the chromosome level (Figure 2)”

I looked at Figure 2. It seems to me for most chromosomes, the gypsy TIPs demonstrate a bias toward centromeric and pericentromeric regions as well as short arms of chr04 and chr10. These regions are known to be rich on gypsy elements. If the authors want to keep this statement, there should be a statistic test of comparison between chromosomal arms and pericentromeric regions.

We agree with Reviewer 3 and addressed this particular point in the discussion.

3. p.6 bottom “Furthermore, we did not find any difference between Gypsy and Copia elements in respect to the distance of insertion from genes (Table 1).

First of all, it is unclear to me whether the authors refer to all elements or just TIPs, so please clarify. Second, I am very much surprised that there is no difference between gypsy and copia. There are multiple studies indicating that copia elements are closer to genes than gypsy elements. I did not find any details about how they calculate the distance. My suggestion is, the authors should double check everything. Particularly, it is important to exclude transposon genes from the current gene set. If they still get the same results, they should discuss why their result is different from previous results.

We performed some statistical analyses and indeed found that there is a significant difference in Gypsy and Copia insertions relative to genes. We therefore modified our manuscript accordingly.

4. p.8 middle “ar at least which may be active in agro since its presence is” should be “or at least ...”.

Done

Reviewer #4 (Remarks to the Author):

*This paper provides a retrotransposon-focused analysis of a population-scale genomics dataset of cultivated rice (*Oryza sativa* L.) recently published by the 3,000 Rice Genome Project. The impressive size of the dataset is unprecedented in crop research, which poses specific computational challenges for genomic analyses, but also promises new discoveries and insights into genomic organisation, evolution, population history etc. Transposable elements (TEs) comprise large fractions of crop genomes which are no longer considered irrelevant in respect to genome organisation and complex regulation of*

gene expression, and consequently, phenotypic variation. It is therefore of high importance to characterise TE profiles and the dynamics of TE evolution in crop species.

Through clearly defined objectives, appropriate methodology and good organisation, this paper presents the following new findings and technical advances:

(I) new software TRACKPOSON designed for detection of TE insertion polymorphisms (TIPs) in large datasets

(II) most of TIPs were found at very low frequencies, which is interpreted as evidence of very recent activity (following domestication)

(III) GWAS did not identify clear associations of nucleotide variants and the TE copy numbers, which was interpreted to suggest TE proliferation as a response to environmental stimuli

(IV) application of the molecular clock principle resulted in dating the split between indica and japonica gene pools long before the beginnings of agriculture, which is interpreted as evidence of two separate domestication processes

In respect to (I), I think that TRACKPOSON offers a clever way how to speed-up the TIPs detection, thereby allowing such analysis in population-scale whole-genome datasets. I don't see why mapping all data onto the genome reference (the first step of alternative methods) could have any advantages for the TIPs detection, therefore I think that removing or reducing this step is safe and should not affect sensitivity or specificity of the detection.

Question: I don't see how TRACKPOSON (or similar methods) could distinguish two insertions of the same class TE in two individuals at nearby locations. E.g., two Hopi retrotransposons are inserted upstream of a gene XY, each insertion occurring in different individuals but very similar (non-identical) location. Are such insertions correctly resolved as unique, or incorrectly as a single event?

At this point, the software can not identify multiple insertions of the same element in the same 10 kb window. We have performed additional analyses reducing window size to 1 kb and did not find significant differences in the results.

(II) The low frequencies of most TIPs are really intriguing and I am inclined to accept authors' interpretation that this implies their origin in agro. However, in my understanding, TIPs do not only result from insertions of new TE copies, but also from removal of the old ones. The removal is recombination dependent and usually results in a single LTR remnant. Can we be sure that the low frequencies of TIPs reported here are not due to a very dynamic TE-removal process? Many older TEs can be completely removed, hence undetected, while others are removed incompletely and persist in the population, albeit in low frequencies. Does the detection of TIPs associated with single LTRs and complete TEs have different sensitivities? Can you distinguish TIPs associated with full TEs from those associated with single LTRs?

Since the libraries used for the 3k project were standard 300bp insert libraries, most selected reads map onto the LTR (because the mate pair must map onto the flanking sequence). We can not exclude the possibility that the TIPs that we identified are from full elements vs solo-LTRs. However, solo-LTR should be detected with the same efficiency as full length elements.

(III) I think this is technically good. However, I find the wording of the conclusions a bit confusing. The authors write: "we also show that the activation of transposition in rice is probably triggered by an external stimulus, rather than by the impediment of a silencing pathway". In my understanding, the authors are trying to differentiate between genetic mutations in genes involved in the epigenetic modification pathways (methyltransferases or RNA-directed DNA methylation pathway) – and external stimuli, i.e. biotic and abiotic stresses. However, it appears that the environmental stresses do not cause demethylation directly – but rather indirectly by modulating the DNA methylation pathways (Downen et al. 2012; Luna et al. 2012; Yu et al. 2013). This process may be governed by salicylic acid-mediated stress response. Hence, the "external stimulus", "impediment of a silencing pathway" and demethylation are all interconnected and I find it a bit confusing to put them in contrast. I suggest to rephrase.

We rephrased.

(IV) This part of the manuscript does not bring much novelty, but can be significantly improved by additional analysis (see below). First, the conclusion that the genomes of indica and japonica are separated by several hundred thousand years is old (Ma & Bennetzen 2004; Vitte et al. 2004 – the same senior author as in this manuscript) and generally accepted. However, it is becoming clear that *O. sativa* includes a third group – aus – and it is not yet widely accepted that aus and indica represent two gene pools with separate origins. The 3,000 RGP dataset contains >200 aus accessions. Expanding

the analyses to include this group could help resolve the issue of aus origin. I would be very excited to see the dating of the indica-aus split, and this could have very high impact.

We have done additional analyses that include Aus/Boro group and show that indeed, this thirs group originates from a distinct, third domestication event.

Secondly, the debate of indica-japonica domestication has moved away from the genome-wide comparisons. The observation that the “genomic backgrounds” (the backgrounds of the domestication genes) in indica and japonica are different has been supported by solid evidence. Currently – as the authors rightly noted – the question of domestication revolves around the genealogical origin of the domestication genes. Proponents of the introgression hypothesis claim that a few genes critical for the domestication phenotype were transferred from japonica to indica by introgression, and hence the domestication phenotype originated in a single domestication event and was later transferred into proto-domesticated or wild indica. Although I do not agree with that model and see it as an ad hoc hypothesis to save the single domestication model from being falsified, it is still a plausible scenario. Importantly, if it is correct, then the observation of indica-japonica differences at the genome-wide level does not prove independent domestications. Instead, focusing on the regions surrounding the domestication genes is essential to resolve this problem; however, the authors say that the TIPs do not have sufficient densities for such comparisons. Taking all these points into account, the claim “rice originated from two distinct domestication events” is unsubstantiated and should be changed to something like “rice originated from at least two distinct gene pools”.

Rephrased, as suggested by another referee.

In relation to this, the authors also say “These two gene pools must have split several hundreds of thousand years ago, possibly due to the rise of the Himalayas.” I find this idea geographically confused. The Himalayas split the Asian continent into a southern part (the Indian subcontinent) and the northern part. However, wild rice does not grow north of the Himalayas, and the indica-japonica gene pool split is rather longitudinal – with the progenitor of japonica found in southern and eastern China, and the progenitors of aus and indica in the Indian subcontinent and Indochina (Civan et al. 2015; Civan and Brown 2018).

The authors also say “Alternatively, one could also argue that the domesticated alleles have been present in the populations of the wild progenitor of rice long before the split of the two gene pools that gave rise to the domesticated forms and then selected for two times independently.” This was first suggested in Civan and Brown 2017 and later confirmed in Civan and Brown 2018; and it is also consistent with the results of Wang et al. 2018. I think these works should be referenced.

These authors have been cited.

Other questions:

The authors say that they analysed 1067 traditional varieties included in the 3,000 genomes. However, the identities of the 1067 traditional varieties are not given, and I cannot find such characterisation in the online resources of the 3k RGP dataset. Can you provide a list?

The information is in the description of the 3k varieties. I can provide to the reviewer since he signed his review.

Peter Civan, 14. 05. 2018, Manchester, UK

References:

*Downen et al. 2012. Proc Natl Acad Sci U S A. 109:E2183–E2191.
Luna et al. 2012. Plant Physiol 158:844–853.
Yu et al. 2013. Proc Natl Acad Sci U S A. 110:2389–2394.
Ma & Bennetzen 2004. Proc Natl Acad Sci U S A.101:12404–10.
Vitte et al. 2004. Mol Gen Genom. 272:504–11.
Civan et al. 2015. Nat plants 1:15164.*

Civan & Brown 2017. *Genet Resour Crop Evol.* 64:1125–1132.
Civan & Brown 2018. *BMC Evol Biol.* 18:57.
Wang et al. 2018. *Nature.* 557, pages43–49

Reviewers' comments:

Reviewer #1 (Remarks to the Author):

The authors addressed my main criticism, the missing benchmarking of Trackposon, by generating a long-read dataset (Nanopore seq) for one accession. This is a strong addition to the manuscript and provides a good estimate of sensitivity and specificity of Trackposon. The Nanopore data is also a nice resource for testing other TE or SV methods for rice. Furthermore a comparison with the tool jitterbug was performed, with favourable results for Trackposon.

While I agree with the presented solution for benchmarking, I am not happy with the detail provided in the Methods section. It is unclear how Nanopore reads are used to identify novel insertion sites. To my understanding the Method section only explains how Nanopore reads containing TE sequence are identified. This needs to be better explained.

Furthermore, the use of the jitterbug tool is not explained at all in the Methods section. Which alignment tool was used to create the BAM files? Which parameters and input files were used to run jitterbug?

And as the authors state that the read length used for the rice 3000k project is not good for jitterbug's split-read approach, why compare to this tool and not to one that solely uses PEM?

In summary, I very much like the addition of Nanopore data for benchmarking of TE insertion detection tools. But the benchmark procedure has to be explained with enough detail to assess the validity of the comparison.

Reviewer #2 (Remarks to the Author):

The responses provided by the authors are satisfactory overall. However, further clarification is required in places:

1) Page 6: "We first tested TRACKPOSON on a rice mutant for which TIPs had been previously characterized by using our previous PEM-based software [39] and wet-lab validated by PCR amplification and sequencing (Supplemental data 1). All TE insertions were detected, which suggests that TRACKPOSON is a robust detection procedure with very limited FDR...". The second sentence is ambiguous and it is not clear if the analysis refers to false positives or false negatives.

2) Page 6: "In order to check for the efficiency of this method..." Replace "efficiency" with "performance". Why was sensitivity tested using only three families? This is particularly important to know, given the large differences of performance observed between these three families.

3) Page 8: "We chose a sample of families that are representative of the diversity found in this accession in terms of number of repeats, from very moderately (e.g. Tos17 with two complete copies in the reference genome) to highly repeated families (e.g. Houba with 150 complete copies). Therefore, we consider that the features revealed by this sample of retrotransposons should be representative of the complete retrotranspositional landscape of rice genome shaped by the whole 300 LTR-retrotransposon families [16]." The last sentence does not provide any hard evidence supporting the claim that the 31 retrotransposon families studied are representative of the complete retrotranspositional landscape of the rice genome. Indeed, it is entirely possible that some retrotransposon families are not represented by any full-length element in the reference genome while being active in other genomes.

4) Page 8: The authors should describe in more detail the analysis that now leads them to find significant differences in the position of Gypsy and Copia elements relative to genes.

5) Page 11: PCoA analysis. Figure 5A and 5B are still impossible to read as the two first axes are not labeled, (the percentage of variance they explain should be indicated as well as the scales). The meaning of the colored lines and circles is not clear.

6) Page 11: "These TIPs were then classified into seven distinct categories : 1), 2) and 3) : Indica, Japonica and Aus/Boro specific insertions respectively, i.e. present in at most 10 % of the other varietal groups present in all traditional varieties, regardless their varietal group...": How can insertions be specific to a group and be present in another one?

7) Page 11: Figure 6. Violin plots still go above 100%., because of the low number of data in each category. Box plots would therefore be more appropriate here.

8) Page 11: Why is the rate of the molecular clock not mentioned anymore? Why this rate measured genome-wide? This may not be appropriate, given retrotransposition is error-prone and therefore leads to more rapid accumulation of mutations between retrotransposed copies and biases the estimate of their age when based solely on percentage of identity.

9) Page 13: Is it really more parsimonious to suggest three independent domestication events and the replacement at several domestication loci of two of the three (independent) alleles by the third one rather than one domestication event and introgression of a single domesticated allele at each locus into the other two genomes with distinct TE landscapes?

Reviewer #3 (Remarks to the Author):

Comments to authors

Review for "Retrotranspositional landscape of Asian rice (*Oryza sativa*) revealed by 3,000 genomes"

This is a revision. The authors indeed addressed most of my comments, so I am happy with that. There is no doubt that the manuscript has significantly improved. However, there are some new issues arisen with the revision. Please see below.

Major comment – rice domestication history

In the first version of the manuscript, the authors proposed that there are two distinct domestication events, but now they argue there are three events. This made me feel it is quite subjective. According to Garris et al. (Genetic structure and diversity in *Oryza sativa* L Genetics, 169 (2005), pp. 1631-1638), there are five subpopulations among rice cultivars. In addition to the three subpopulations mentioned in the manuscript, Japonica rice could be further divided into tropical Japonica, temperate Japonica, and aromatic. I think the author should test whether the splitting of these three groups was before or after domestication. If it was before domestication, are we going to suggest that there were five distinct domestication events?

This is the first comprehensive study about the activity of LTR elements in a large rice population, so I anticipated that the current Figure 6 would be heavily cited. As a result, I would like it to be as accurate as possible. My understanding about Figure 6B is that the authors are suggesting that the ancestor of rice first split into japonica and the ancestor of indica and aus, and then the second split was between indica and aus. This is fine with me if they are going to stick with the three domestication events hypothesis. However, I do have questions where the split time came from. Apparently, the authors used the common insertions between indica and japonica to estimate the first split. However, if the authors consider that indica and aus share a common ancestor, I think the common insertions among three populations should be used for estimation.

In fact, I consider the age of the common insertions is the upper limit of the splitting and the age of population specific insertions represents the lower limit of the splitting. Obviously, the splitting time should be between the two limits. Moreover, I would guess there is an age overlap between the common and specific insertions. So my suggestion is, for the first split, they should make a

distribution (age vs frequency) of common insertions in the three population, and the other distribution is that for all population specific insertions, and the center of the overlap should be used to estimate the splitting time. Similarly, for the second split, they should make a distribution about the common insertions between indica and aus, and distribution of specific insertions in indica and aus, then looking for overlap for the possible splitting time.

Minor comment – recent or old activity?

p.9 first paragraph “Finally, for a few families (e.g., Dasheng), most insertions were found at high frequencies, which suggests more ancient activity.”

However, on page 11, it was stated “For this, we first identified all insertions of full elements in Nipponbare (for Japonica), IR8 (for Indica) and N22 (for Aus/Boro) high quality genome assemblies [22] for the 9 TE families showing the highest number of TIPS, i.e. Dagul, Dasheng, Hopi, Houba, Osr37, Rire2, Rire3, RN215 and Poprice”.

In addition, on table 1, Dasheng was marked as “recent” activity.

First of all, I think the first statement (ancient activity) is contradictory to the data on Table 1 as well as what was suggested by the second statement.

Second, allele frequency is not only determined by amplification time, but also the selection pressure on the insertions. For example, if an insertion is deleterious but not fatal, it could be present in the population for a while but never achieve high frequency.

Anyway, please make the statements and data consistent with each other.

Reviewer #4 (Remarks to the Author):

My questions from the previous round of reviews were answered satisfactorily and all points raised by me were resolved in this revised version. I particularly appreciate that the authors expanded their analyses onto the aus group, and I find their results very interesting. This paper provides the first observation-based evidence and timing of aus—indica divergence (after the modelling-based approach by Choi et al. 2017. *Mol Biol Evol* 34:969) and provides clear evidence that the lineage leading to aus was separated from the other lineages long before domestication.

In my opinion, several minor issues (below) can be raised in this revised version, but I believe they can be easily addressed/resolved and the quality of the manuscript improved further.

1. Would it be possible to prepare a supplementary figure/diagram explaining the genomic paleontology approach schematically?

2. You give the %identity medians of the group-specific TE insertions, and then the %identity medians of the communal TE insertions. Then you say “the distribution median of Indica-Japonica common insertions are 97.9 % (~800,000 years ago), showing that the two gene pools of *O. rufipogon* from which both cultivated types originated were separated long before the origin of agriculture). The split of the Indica/Aus lineages appears to be more recent with a median at 98.6 % (~540,000 years ago)”. However, in my view, the shared TE insertions medians do not tell us when the lineages split/diverged. Instead, they tell us the time when the two lineages were for sure still together (and encountered a transpositional burst). Saying that indica and japonica diverged 0.8 mya may be incorrect, because the median suggests that they were in fact together at that timepoint. Instead, it is the group-specific medians that tell us when the lineages were for sure separated (and encountered transpositional bursts). Therefore, I would say that the divergence time should be placed in between the group-specific medians and the communal medians. Interestingly, the aus-specific median is at 380ky ago, earlier in time compared to the indica-specific and japonica-specific medians. Also, the last three violin plots are difficult to reconcile with the suggested phylogeny (or rather the “population tree”). The 6th plot suggests aus+indica encountered a communal transpositional burst 540ky ago, but the 5th and 7th plot suggest that indica and aus were separate lineages before then (because they have different transpositional bursts communal with japonica). How do you explain this conflict?

3. You say that the group-specific medians can be used as the “upper limit of the time of divergence”. I know this is a matter of perspective, but I would call it the lower limit (as the

number of years is lower than 800ky).

4. As mentioned earlier, Choi et al. 2017 attempted to date the divergence of indica, japonica and aus lineages, using a coalescent modelling approach. Their divergence times are much younger. Although their methodology has been criticized (Civan et al. 2018. BMC Evol Biol 18:57), can you compare your results to Choi et al. 2017?

5. I suggest providing brief explanation to the violin plots in the figure caption. What do the white bars, vertical lines and colored areas represent?

6. The Supplementary Figure 2 shows $\log_2(\text{depth of coverage})$ plotted against $\log_2(\text{Hopi insertions number})$. Based on real numbers from the tables, it seems to me that you used $\ln(x)$, not $\log_2(x)$ for data transformation.

7. I would like to go back to your conclusion that many detected transpositions have occurred in agro, based on the observation of many low-frequency TIPs (found in 1-2 landraces). I agree with this conclusion, but it is worth mentioning that the frequency estimates depend on the sensitivity of TRACKPOSON, which is about 81% for Hopi. In fact, the supplementary figure 2 nicely shows that even at the threshold=2, the number of TEs detected clearly depends on the depth of coverage. Hence, if the depth of coverage was $>30x$ for all accessions, is it possible that much fewer TIPs would have the extremely low frequencies?

8. You say that "The higher TE density usually found in pericentromeric regions in plant genomes could thus result from retention of TEs in addition to an insertional bias in these regions for Gypsy elements." I thought that no mechanism of preferential insertion into pericentromeric regions has been described so far (though, I may be wrong). In my understanding, the distribution bias (not insertional bias) is solely due to TEs being preferentially removed from telomeric (or highly recombining) regions, since the removal of TEs is recombination dependent.

9. In the sentence "These TIPs were then classified into seven distinct categories : 1), 2) and 3) : Indica, Japonica and Aus/Boro specific insertions respectively, i.e. present in at most 10 % of the other varietal groups [present in all traditional varieties, regardless their varietal group]; 4)..." – I think the part I put in the brackets has been pasted there by mistake.

10. TRACKPOSON pipeline – since the detection depends on the number of supporting reads – were the raw reads deduplicated prior to analyses? (e.g. by tally from the reaper program)

11. I find it difficult to interpret the PCoA.

12. I suggest rephrasing at one or two places in the second-to-last paragraph of the discussion. E.g., *O. rufipogon* is not really found "all across the continent" (e.g. it is not found in Siberia), and the phrase "both sides of the Himalayas" is still confusing to me. In the next sentence, the verb "must" is not the best, since an alternative is offered a sentence later.

Peter Civan,

17. Sept. 2018

Reviewers' comments:

Reviewer #1 (Remarks to the Author):

The authors addressed my main criticism, the missing benchmarking of Trackposon, by generating a long-read dataset (Nanopore seq) for one accession. This is a strong addition to the manuscript and provides a good estimate of sensitivity and specificity of Trackposon. The Nanopore data is also a nice resource for testing other TE or SV methods for rice. Furthermore a comparison with the tool jitterbug was performed, with favourable results for Trackposon.

While I agree with the presented solution for benchmarking, I am not happy with the detail provided in the Methods section. It is unclear how Nanopore reads are used to identify novel insertion sites. To my understanding the Method section only explains how Nanopore reads containing TE sequence are identified. This needs to be better explained.

We completed the methods section : we now explain the procedure we followed for the mapping of the TE insertions :

« Validation TIPs with Nanopore sequencing :

We resequenced one Indica rice variety - ie, IRIS-313-11419 using the Nanopore long-read technology. The library was prepared using the 1D kit according to the manufacturer's instructions. One R9.4 flowcell was used. After basecalling of long reads with Albacore Oxford Nanopore software (to convert fast5 files in fasta format), a blast database was created. We performed a blastn of the TE families against this Nanopore database, with evaluate 1e-50 and penalties for open and extend gap equal to 0. »

was changed to :

« Validation TIPs with Nanopore sequencing :

We resequenced one Indica rice variety - ie, IRIS-313-11419 using the Nanopore long-read technology. The library was prepared using the 1D kit according to the manufacturer's instructions. One R9.4 flowcell was used. After basecalling of long reads with Albacore Oxford Nanopore software (to convert fast5 files in fasta format), a blast database was created. We performed a blastn of the TE families against this Nanopore database, with evaluate 1e-50 and penalties for open and extend gap equal to 0.

Only the reads with a HSPs corresponding to maximum of 80% of their length were kept. With this filtering, we eliminated the reads corresponding only to a TE. All the reads were validated by hand by dotter and NCBI blast against the rice reference genome IRGSP1.0. For each read thus selected, 300 bp of sequences flanking the insertion were used as query for blastn search against the IRGSP 1.0 rice pseudomolecules with 1e-50 e value threshold. This allowed unambiguous mapping of the TE insertions.

The sequencing data is available at http://gamay.univ-perp.fr/~Panaudlab/Nanopore_read_IRIS-313-11419_136184.fasta.tar.gz »

Furthermore, the use of the jitterbug tool is not explained at all in the Methods section. Which alignment tool was used to create the BAM files? Which parameters and input files were used to run jitterbug?

We added this information in the method section : the following paragraph was added :

«Benchmarking with JITTERBUG software :

The Illumina sequencing data of Mutha Samba accession was used for TE detection using JITTERBUG software. First, all paired reads were mapped onto the rice reference genome (IRGSP 1.0) with Bowtie2 (v. 2.2.0) in very-sensitive mode. Then TE insertions were detected with JITTERBUG using the default parameters. «

And as the authors state that the read length used for the rice 3000k project is not good for jitterbug's split-read approach, why compare to this tool and not to one that solely uses PEM?

Jitterbug when published was benchmark against Retroseq which uses PEM. Reference added + we changed :

« As mentioned in the introduction, several softwares for TIPs detection are available, but we anticipated that the computation time needed for the complete characterization of large datasets would exceed the resources allocated in a single project in most genomic facilities, making them inappropriate for routine analyses in the future, as one could anticipate that population genomic data will dramatically increase in the coming years. To illustrate this, we ran both TRACKPOSON and JITTERBUG [27], a recently published TIPs detection software using Paired-End-Mapping and Split-Read methods, on the same sequence dataset. First, both sensitivity and specificity were compared for the detection of insertions of the three same families as above (*i.e.* *Tos17*, *Karma* and *Hopi*). JITTERBUG detected 100% of the *Karma* insertions but failed to detect 6 of the 8 *Tos17* insertions and detected one false positive insertion for this family. For *Hopi*, the program detected 85 of the 581 insertions present in the genome of the test variety. The low sensitivity of JITTERBUG with the 3,000 rice genomes dataset (14,6 %) is probably caused by the short length of sequencing reads (83 nt) that prevent reliable

detection of insertions using split-read method. The computation time needed by both softwares was also compared : it took 80 min and 200 min to detect TIPS of the 3 families in the test variety for TRACKPOSON and JITTERBUG, respectively. This demonstrates the efficiency of TRACKPOSON in terms of computing time. “

by :

“ As mentioned in the introduction, several softwares for TIPS detection are available, but we anticipated that the computation time needed for the complete characterization of large datasets would exceed the resources allocated in a single project in most genomic facilities, making them inappropriate for routine analyses in the future, as one could anticipate that population genomic data will dramatically increase in the coming years. To illustrate this, we ran both TRACKPOSON and JITTERBUG [27], a TIPS detection software using both Paired-End-Mapping and Split-Read methods, on the same sequence dataset. JITTERBUG was chosen because it was recently published and proven to be superior to RETROSEQ, another recently published software for TE detection [42]. First, both sensitivity and specificity were compared for the detection of insertions of the three same families as above (*i.e.* *Tos17*, *Karma* and *Hopi*). JITTERBUG detected 100% of the *Karma* insertions but failed to detect 6 of the 8 *Tos17* insertions and detected one false positive insertion for this family. For *Hopi*, the program detected 85 of the 581 insertions present in the genome of the test variety. The low sensitivity of JITTERBUG with the 3,000 rice genomes dataset (14,6 %) is probably caused by the short length of sequencing reads (83 nt) that prevent reliable detection of insertions using split-read method. The computation time needed by both softwares was also compared : it took 80 min and 200 min to detect TIPS of the 3 families in the test variety for TRACKPOSON and JITTERBUG, respectively. This demonstrates the efficiency of TRACKPOSON in terms of computing time. “

In summary, I very much like the addition of Nanopore data for benchmarking of TE insertion detection tools. But the benchmark procedure has to be explained with enough detail to assess the validity of the comparison.

Reviewer #2 (Remarks to the Author):

The responses provided by the authors are satisfactory overall. However, further clarification is required in places:

1) Page 6: “We first tested TRACKPOSON on a rice mutant for which TIPS had been previously characterized by using our previous PEM-based software [39] and wet-lab validated by PCR amplification and sequencing (Supplemental data 1). All TE insertions were detected, which suggests that TRACKPOSON is a robust detection procedure with very limited FDR...”. The second sentence is ambiguous and it is not clear if the analysis refers to false positives or false negatives.

We corrected the text as follows (p6) :

“ All TE insertions were detected, which suggests that TRACKPOSON is a robust detection procedure with very limited FDR, given that sufficient genome coverage has been achieved (the mutant was sequenced at 30x depth). ”

was replaced by :

“ All TE insertions were detected, which suggests that TRACKPOSON is a robust detection procedure with **high sensitivity**, given that sufficient genome coverage has been achieved (the mutant was sequenced at 30x depth).”

2) Page 6: “In order to check for the efficiency of this method...” Replace “efficiency” with “performance”. Why was sensitivity tested using only three families? This is particularly important to know, given the large differences of performance observed between these three families.

Page 6, the following paragraph

“In order to check for the efficiency of this method we resequenced one rice accession (Mutha Samba::IRGC 49924-2) that was originally sequenced in the 3,000 genome collection at a 8.3x depth, *i.e.* among the lowest in the dataset. For this variety, TRACKPOSON identified 501, 8 and 9 insertions for *Hopi*, *Tos17* and *Karma*, respectively. We used Nanopore long-read sequencing technology (Minion device). One flowcell was used and produced 4,77 Gbp of sequence (11x genome coverage). We manually checked for the presence of TIPS of *Hopi*, *Tos17* and *Karma*, from the reads thus generated [40]. 100 % of the insertions of both *Tos17* and *Karma* identified by TRACKPOSON were validated. In the case of *Hopi*, 473 insertions were validated. These results show that the specificity of TRACKPOSON is ~ 94.5 %. The rate of false negatives (sensitivity) was also estimated : while there was a 100% overlap between the insertions detected by TRACKPOSON and those detected using Nanopore reads for both *Tos17* and *Karma*, which is indicative of 100% sensitivity for these two families, we detected a total of 581 *Hopi* insertions from the Nanopore dataset, which corresponds to a drop of sensitivity to 81% for this family. However, the mappability of a complex genome such as that of rice with small reads obtained on Illumina platforms is expected to not be 100 % because of the presence of repeats that impede unambiguous mapping at unique sites [41]. In the case of rice, the mappability of 100bp windows at a $1e^{-20}$ threshold (*i.e.* that used for the second step of the detection, see Methods section) was estimated to be 63.5 %, on average. This estimation fits with previous estimates of the repeat content of *Japonica* rice genome (*i.e.* ~ 40% [15]). We determined the mappability of the regions where we detected *Hopi* insertions with Nanopore reads. We found an average of 58% for all insertions. However, the insertions found only in the nanopore data (and not detected by TRACKPOSON) have a significant lower mappability of 42%. This may explain the lower sensitivity of our software for this particular family.”

was replaced by :

“ In order to check for the performance of this method, we resequenced one rice accession (Mutha Samba::IRGC 49924-2) that was originally sequenced in the 3,000 genome collection at a 8.3x depth, i.e. among the lowest coverage in the dataset. We used Nanopore long-read sequencing technology (Minion device). One flowcell was used and produced 4,77 Gbp of sequence (11x genome coverage). We manually checked for the presence of TIPS of Hopi, Tos17 and Karma, from the reads thus generated [40]. We chose these three families because they are representative of the diversity of the retrotransposons found in the reference rice genome, i.e. a low (Tos17) and a high (Hopi) copy number LTR-retrotransposon families and a moderately repeated LINE family (Karma). With this data, TRACKPOSON identified 501, 8 and 9 insertions of Hopi, Tos17 and Karma, in Mutha Samba cultivar, respectively. 100 % of the insertions of both Tos17 and Karma identified by TRACKPOSON were validated. In the case of Hopi, 473 insertions were validated. These results show that the specificity of TRACKPOSON is ~ 94.5 %. The rate of false negatives (sensitivity) was also estimated : while there was a 100% overlap between the insertions detected by TRACKPOSON and those detected using Nanopore reads for both Tos17 and Karma, which is indicative of 100% sensitivity for these two families, we detected a total of 581 Hopi insertions from the Nanopore dataset, which corresponds to a drop of sensitivity to 81% for this family. However, the mappability of a complex genome such as that of rice with small reads obtained on Illumina platforms is expected to not be 100 % because of the presence of repeats that impede unambiguous mapping at unique sites [41]. In the case of rice, the mappability of 100bp windows at a $1e^{-20}$ threshold (i.e. that used for the second step of the detection, see Methods section) was estimated to be 63.5 %, on average. This estimation fits with previous estimates of the repeat content of Japonica rice genome (i.e. ~ 40% [15]). We determined the mappability of the regions where we detected Hopi insertions with Nanopore reads. We found an average of 58% for all insertions. However, the insertions found only in the nanopore data (and not detected by TRACKPOSON) have a significant lower mappability of 42%. This may explain the lower sensitivity of our software for this particular family.”

3) Page 8: “We chose a sample of families that are representative of the diversity found in this accession in terms of number of repeats, from very moderately (e.g. Tos17 with two complete copies in the reference genome) to highly repeated families (e.g. Houba with 150 complete copies). Therefore, we consider that the features revealed by this sample of retrotransposons should be representative of the complete retrotranspositional landscape of rice genome shaped by the whole 300 LTR-retrotransposon families [16].” The last sentence does not provide any hard evidence supporting the claim that the 31 retrotransposon families studied are representative of the complete retrotranspositional landscape of the rice genome. Indeed, it is entirely possible that some retrotransposon families are not represented by any full-length element in the reference genome while being active in other genomes.

We added a comment on this. In the paragraph above, we mentioned :

« We manually checked for the presence of TIPS of Hopi, Tos17 and Karma, from the reads thus generated [40]. We chose these three families because they are representative of the diversity of the retrotransposons found in the reference rice genome, i.e. a low (Tos17) and a high (Hopi) copy number LTR-retrotransposon families and a moderately repeated LINE family (Karma)”

4) Page 8: The authors should describe in more detail the analysis that now leads them to find significant differences in the position of Gypsy and Copia elements relative to genes.

The description is in the supplemental data 3:

« We performed a Welch t-test between these two distributions with p.value of $1.6e^{-07}$. The Copia and Gypsy distribution are significantly different and the Copia TIPS are closer to the genes. »

5) Page 11: PCoA analysis. Figure 5A and 5B are still impossible to read as the two first axes are not labeled, (the percentage of variance they explain should be indicated as well as the scales). The meaning of the colored lines and circles is not clear.

We now changed the figure and used DAPC instead of PcoA.

6) Page 11: “These TIPS were then classified into seven distinct categories : 1), 2) and 3) : Indica, Japonica and Aus/Boro specific insertions respectively, i.e. present in at most 10 % of the other varietal groups present in all traditional varieties, regardless their varietal group...”: How can insertions be specific to a group and be present in another one?

Sentence removed.

7) Page 11: Figure 6. Violin plots still go above 100%., because of the low number of data in each category. Box plots would therefore be more appropriate here.

We changed the figure. Box plots are now used.

8) Page 11: Why is the rate of the molecular clock not mentioned anymore? Why this rate measured genome-wide? This may not be appropriate, given retrotransposition is error-prone and therefore leads to more rapid accumulation of mutations between retrotransposed copies and biases the estimate of their age when based solely on percentage of identity.

We don't understand this remark. The molecular clock rate is still in the text....

The molecular clock commonly used for retrotransposon paleontology is the one published by Ma and Bennetzen in 2004 (Proc Natl Acad Sci U S A. 101(34):12404-10.PNAS). The authors have used a comparative genomics approach to unravel the dynamics of genome expansions and contractions in *Oryza* genus. Below is the abstract of this paper :

« Abstract

By employing the nuclear DNA of the African rice *Oryza glaberrima* as a reference genome, the timing, natures, mechanisms, and specificities of recent sequence evolution in the *indica* and *japonica* subspecies of *Oryza sativa* were identified. The data indicate that the genome sizes of both *indica* and *japonica* have increased substantially, >2% and >6%, respectively, since their divergence from a common ancestor, mainly because of the amplification of LTR-retrotransposons. However, losses of all classes of DNA sequence through unequal homologous recombination and illegitimate recombination have attenuated the growth of the rice genome. Small deletions have been particularly frequent throughout the genome. In >1 Mb of orthologous regions that we analyzed, no cases of complete gene acquisition or loss from either *indica* or *japonica* were found, nor was any example of precise transposon excision detected. The sequences between genes were observed to have a very high rate of divergence, indicating a molecular clock for transposable elements that is at least 2-fold more rapid than synonymous base substitutions within genes. We found that regions prone to frequent insertions and deletions also exhibit higher levels of point mutation. These results indicate a highly dynamic rice genome with competing processes for the generation and removal of genetic variation. »

At that time, the authors used 14 randomly chosen loci shown to be orthologous between *Indica* (Gla4 variety) and *japonica* (Nipponbare variety). They then amplified and sequenced orthologous sequences from *O. glaberrima*, thus estimating the substitution rate of both genic and TE-related sequences, confirming one one hand that genic molecular clock is $\sim 6.5 \times 10^{-9}$ subst/site/year and estimating on the other that LTR-retrotransposon clock runs \sim twice faster, i.e. 1.3×10^{-9} subst/site/year.

We are well aware that the use of molecular clock to date TE insertion events has been debated. However, in order to reach the conclusion that only one domestication event occurred, one should hypothesize that the molecular clock of LTR-retrotransposons of rice runs ~ 100 x faster (i.e. 6×10^{-7} substitutions/site/year, in order to explain that *Indica*-specific insertions would accumulate 0,6 % divergence in 10,000 years). If this is the case, then this is a major discovery.

9) Page 13: Is it really more parsimonious to suggest three independent domestication events and the replacement at several domestication loci of two of the three (independent) alleles by the third one rather than one domestication event and introgression of a single domesticated allele at each locus into the other two genomes with distinct TE landscapes?

We think that it is indeed more parsimonious given the monophyletic origin of *Indica* rice :

If « one domestication event » had occurred followed by « introgression of a single domesticated allele at each locus into the other two genomes with distinct TE landscapes » was the actual scenario for the origin of *Indica* rice, then how to explain that *Indica* rice exhibits a much higher diversity than *japonica* does ?

Reviewer #3 (Remarks to the Author):

Comments to authors

Review for “Retrotranspositional landscape of Asian rice (*Oryza sativa*) revealed by 3,000 genomes”

This is a revision. The authors indeed addressed most of my comments, so I am happy with that. There is no doubt that the manuscript has significantly improved. However, there are some new issues arisen with the revision. Please see below.

Major comment – rice domestication history

In the first version of the manuscript, the authors proposed that there are two distinct domestication events, but now they argue there are three events. This made me feel it is quite subjective. According to Garris et al. (Genetic structure and diversity in *Oryza sativa* L Genetics, 169 (2005), pp. 1631-1638), there are five subpopulations among rice cultivars. In addition to the three subpopulations mentioned in the manuscript, *Japonica* rice could be further divided into tropical *Japonica*, temperate *Japonica*, and aromatic. I think the author should test whether the splitting of these three groups was

before or after domestication. If it was before domestication, are we going to suggest that there were five distinct domestication events?

This is the first comprehensive study about the activity of LTR elements in a large rice population, so I anticipated that the current Figure 6 would be heavily cited. As a result, I would like it to be as accurate as possible. My understanding about Figure 6B is that the authors are suggesting that the ancestor of rice first split into japonica and the ancestor of indica and aus, and then the second split was between indica and aus. This is fine with me if they are going to stick with the three domestication events hypothesis. However, I do have questions where the split time came from.

Apparently, the authors used the common insertions between indica and japonica to estimate the first split. However, if the authors consider that indica and aus share a common ancestor, I think the common insertions among three populations should be used for estimation.

In fact, I consider the age of the common insertions is the upper limit of the splitting and the age of population specific insertions represents the lower limit of the splitting. Obviously, the splitting time should be between the two limits.

Moreover, I would guess there is an age overlap between the common and specific insertions. So my suggestion is, for the first split, they should make a distribution (age vs frequency) of common insertions in the three population, and the other distribution is that for all population specific insertions, and the center of the overlap should be used to estimate the splitting time. Similarly, for the second split, they should make a distribution about the common insertions between indica and aus, and distribution of specific insertions in indica and aus, then looking for overlap for the possible splitting time.

Minor comment – recent or old activity?

p.9 first paragraph “Finally, for a few families (e.g., Dasheng), most insertions were found at high frequencies, which suggests more ancient activity.”

However, on page 11, it was stated “For this, we first identified all insertions of full elements in Nipponbare (for Japonica), IR8 (for Indica) and N22 (for Aus/Boro) high quality genome assemblies [22] for the 9 TE families showing the highest number of TIPs, i.e. Dagul, Dasheng, Hopi, Houba, Osr37, Rire2, Rire3, RN215 and Poprice”.

In addition, on table 1, Dasheng was marked as “recent” activity.

First of all, I think the first statement (ancient activity) is contradictory to the data on Table 1 as well as what was suggested by the second statement.

Second, allele frequency is not only determined by amplification time, but also the selection pressure on the insertions. For example, if an insertion is deleterious but not fatal, it could be present in the population for a while but never achieve high frequency.

Anyway, please make the statements and data consistent with each other.

We thoroughly rewrote the paragraph on domestication. First, we now provide a range of dates for the split of the three gene pool rather than a lower limit :

“We then tentatively dated the origin of the three varietal groups *Japonica*, *Indica* and *Aus/Boro* using a genomic paleontology approach [32]. For this, we first identified all insertions of full elements in Nipponbare (for *Japonica*), IR8 (for *Indica*) and N22 (for *Aus/Boro*) high quality genome assemblies [22] for the 9 TE families showing the highest number of TIPs, i.e. Dagul, Dasheng, Hopi, Houba, Osr37, Rire2, Rire3, RN215 and Poprice (see Methods section for the details of the procedure). These TIPs were then classified into seven distinct categories : 1), 2) and 3) : *Indica*, *Japonica* and *Aus/Boro* specific insertions respectively, i.e. present in at most 10 % of the other varietal groups present in all traditional varieties, regardless their varietal group ; 4) ancestral insertions present in all traditional varieties, regardless their varietal group; 5) insertions that are common between *Japonica* and *Indica* but not present in *Aus/Boro* 6) insertions that are common between *Indica* and *Aus/Boro*, but absent from *Japonica* and 7) insertions that are common between *Japonica* and *Aus* but absent from *Indica*. As previously mentioned, most TIPs are found at low frequency and therefore the insertions used for this analysis represented a small fraction of all TIPs (Figure 6). Each TIP thus identified was dated using a molecular clock of 1.3×10^{-8} substitution/site/year [48]. In total, we successfully dated 1476 TIPs from the seven categories (Figure 6A). As expected, the insertions of the fourth category (i.e., common among all varieties, are more ancient than those belonging to the other categories (*Indica*- and *Japonica*-specific, respectively), which illustrates a split of a common wild ancestor into three distinct gene pools. In addition, the distribution medians for *Indica*, *Japonica* and *Aus/Boro* specific TE insertions are 99.4 % identity (~230,000 years ago) and 99.2 % identity (~310,000 years ago) and 99% identity (~380,000 years ago), respectively. These values represent a peak of transpositional activity after the split of the lineages that gave rise to the three cultivated types and could therefore be used to estimate the upper limit of the time of divergence between the genomes of the wild progenitors of the three groups (Figure 6B). The three values are significantly older than 10,000 years, i.e. the origin of rice cultivation at late Neolithic. Moreover, the distribution median of *Indica*-*Japonica* common insertions are 97.9 % (~800,000 years ago), showing that the two gene pools of *O. rufipogon* from which both cultivated types originated were separated long before the origin of agriculture). The split of the *Indica*/*Aus* lineages appears to be more recent with a median at 98.6 % (~540,000 years ago), albeit still significantly older than the domestication. Therefore, our results strongly suggest that *Indica*, *Japonica* and *Aus/Boro* originated from three distinct domestication events. Unfortunately, the density of TIPs of all seven categories was not sufficient to identify traces of introgression in the vicinity of domestication loci.”

Has been changed to :

“We then tentatively dated the origin of the three varietal groups *Japonica*, *Indica* and *Aus/Boro* using a genomic paleontology approach [32]. For this, we first identified all insertions of full elements in Nipponbare (for *Japonica*), IR8 (for *Indica*) and N22 (for *Aus/Boro*) high quality genome assemblies [22] for the 9 TE families showing the highest number of TIPs, i.e. Dagul, Dasheng, Hopi, Houba, Osr37, Rire2, Rire3, RN215 and Poprice (see Methods section for the details of the procedure). These TIPs were then classified into seven distinct categories : 1), 2) and 3) : *Indica*, *Japonica* and *Aus/Boro* specific insertions respectively ; 4) ancestral insertions present in all traditional varieties, regardless their varietal group; 5) insertions that are common between *Japonica* and *Indica* but not present in *Aus/Boro* 6) insertions that are common between *Indica* and *Aus/Boro*, but absent from *Japonica* and 7) insertions that are common between *Japonica* and *Aus* but absent from *Indica*. As previously mentioned, most TIPs are found at low frequency and

therefore the insertions used for this analysis represented a small fraction of all TIPs (Figure 6). Each TIP thus identified was dated using the method of SanMiguel et al. [49] with a molecular clock of 1.3×10^{-8} substitution/site/year [50]. In total, we successfully dated 1476 TIPs from the seven categories (Figure 6A). As expected, the insertions of the fourth category (i.e., common among all varieties), are more ancient than those belonging to the other categories (*Indica*- and *Japonica*-specific, respectively), which illustrates a split of a common wild ancestor into three distinct gene pools. The distribution medians for *Indica*, *Japonica* and *Aus/Boro* specific TE insertions are 99.4 % identity (~230,000 years ago), 99.2 % identity (~310,000 years ago) and 99% identity (~380,000 years ago), respectively. These values represent a peak of transpositional activity after the split of the lineages that gave rise to the three cultivated types and could therefore be used to estimate the lower limit of the time of divergence between the genomes of the wild progenitors of the three groups (Figure 6B). The three values are significantly older than 10,000 years, i.e. the origin of rice cultivation at late Neolithic. The distribution median of *Indica*-*Japonica* common insertions is 97.9 %, which translates into a date of 800,000 years ago. This confirms that the two gene pools of *O. rufipogon* from which both cultivated types originated were separated long before the origin of agriculture. This value also provides an estimate of the upper limit of the date of divergence between *Indica* and *Japonica* progenitors which, combined with the estimated date of the *Indica*- and *Japonica*-specific categories leads us to propose that the date of the divergence ranges from ~300,000 years ago to 800,000 years ago. The split of the *Indica*/*Aus* lineages appears to be more recent with a median at 98.6 % (~540,000 years ago). Combined with the median age of the *Aus*-specific insertions (see above), our results suggest that the split between the progenitors of *Indica* and *Aus* may have occurred between ~230,000 and ~540,000 years ago. However the median date of insertions that are common between *Japonica* and *Aus* (seventh group) appears to be older (96.4 %, translating into a date of 1,4 Mya) than the median date of insertions that are common between *Japonica* and *Indica* (estimated above at 800,000 years ago). The much smaller sample of *Aus* varieties in the 1,067 traditional cultivar's dataset (84 accessions) may explain this discrepancy. Alternatively it could originate from demographic history of the populations of the rice progenitor *O. rufipogon*. Further analyses may help clarifying this last point.

Our results, synthesized in figure 6b strongly suggest that *Indica*, *Japonica* and *Aus/Boro* originated from three distinct domestication events. Unfortunately, the density of TIPs of all seven categories was not sufficient to identify traces of introgression in the vicinity of domestication loci and therefore can not solve the paradox of the presence of a single allele for these loci [34,35,36]. “

We modified the text : we replaced ancient by continuous.

Reviewer #4 (Remarks to the Author):

My questions from the previous round of reviews were answered satisfactorily and all points raised by me were resolved in this revised version. I particularly appreciate that the authors expanded their analyses onto the aus group, and I find their results very interesting. This paper provides the first observation-based evidence and timing of aus—indica divergence (after the modelling-based approach by Choi et al. 2017. Mol Biol Evol 34:969) and provides clear evidence that the lineage leading to aus was separated from the other lineages long before domestication.

In my opinion, several minor issues (below) can be raised in this revised version, but I believe they can be easily addressed/resolved and the quality of the manuscript improved further.

1. Would it be possible to prepare a supplementary figure/diagram explaining the genomic paleontology approach schematically?

We did not add any supplemental data but instead cited the publication of SanMiguel et al. Nature genetics 1998 describing the principle of genomic paleontology.

2. You give the %identity medians of the group-specific TE insertions, and then the %identity medians of the communal TE insertions. Then you say "the distribution median of Indica-Japonica common insertions are 97.9 % (~800,000 years ago), showing that the two gene pools of O. rufipogon from which both cultivated types originated were separated long before the origin of agriculture). The split of the Indica/Aus lineages appears to be more recent with a median at 98.6 % (~540,000 years ago)". However, in my view, the shared TE insertions medians do not tell us when the lineages split/diverged. Instead, they tell us the time when the two lineages were for sure still together (and encountered a transpositional burst). Saying that indica and japonica diverged 0.8 mya may be incorrect, because the median suggests that they were in fact together at that timepoint. Instead, it is the group-specific medians that tell us when the lineages were for sure separated (and encountered transpositional bursts). Therefore, I would say that the divergence time should be placed in between the group-specific medians and the communal medians. Interestingly, the aus-specific median is at 380ky ago, earlier in time compared to the indica-specific and japonica-specific medians. Also, the last three violin plots are difficult to reconcile with the suggested phylogeny (or rather the "population tree"). The 6th plot suggests aus+indica encountered a communal transpositional burst 540ky ago, but the 5th and 7th plot suggest that indica and aus were separate lineages before then (because they have different transpositional bursts communal with japonica). How do you explain this conflict?

We thoroughly rewrote the paragraph on domestication. First, we now provide a range of dates for the split of the three gene pool rather than a lower limit (see the answer to reviewer 3). We also mention the conflict, not being able to explain it. Maybe the small sample size of Aus accessions is at the origin of skewed results. Maybe this conflict results from differences in demographic dynamics of populations of rice progenitors...

“However the median date of insertions that are common between Japonica and Aus (seventh group) appears to be older (96,4 %, translating into a date of 1,4 Mya) than the median date of insertions that are common between Japonica and Indica (estimated above at 800,000 years ago). The much smaller sample of Aus varieties in the 1,067 traditional cultivar’s dataset (84 accessions) may explain this discrepancy. Alternatively it could originate from demographic history of the populations of the rice progenitor *O. rufipogon*. Further analyses may help clarifying this last point. “

3. You say that the group-specific medians can be used as the "upper limit of the time of divergence". I know this is a matter of perspective, but I would call it the lower limit (as the number of years is lower than 800ky).

Corrected, even if younger dates in evolution are usually referred to as upper (eg upper neolithic = late neolithic)...

4. As mentioned earlier, Choi et al. 2017 attempted to date the divergence of indica, japonica and aus lineages, using a coalescent modelling approach. Their divergence times are much younger. Although their methodology has been criticized (Civan et al. 2018. BMC Evol Biol 18:57), can you compare your results to Choi et al. 2017?

We did not compare our results to that of Choi et al. 2017.

5. I suggest providing brief explanation to the violin plots in the figure caption. What do the white bars, vertical lines and colored areas represent?

We now use boxplots

6. The Supplementary Figure 2 shows $\log_2(\text{depth of coverage})$ plotted against $\log_2(\text{Hopi insertions number})$. Based on real numbers from the tables, it seems to me that you used $\ln(x)$, not $\log_2(x)$ for data transformation.

Corrected.

7. I would like to go back to your conclusion that many detected transpositions have occurred in agro, based on the observation of many low-frequency TIPs (found in 1-2 landraces). I agree with this conclusion, but it is worth mentioning that the frequency estimates depend on the sensitivity of TRACKPOSON, which is about 81% for Hopi. In fact, the supplementary figure 2 nicely shows that even at the threshold=2, the number of TEs detected clearly depends on the depth of coverage. Hence, if the depth of coverage was $>30x$ for all accessions, is it possible that much fewer TIPs would have the extremely low frequencies?

We added some comments on this in the results section :

“ The level of polymorphism generated by the 32 retrotransposon families was assessed by using the frequency of insertions found in the 1,067 traditional varieties (Figure 3). Surprisingly, a large portion of TIPs are specific to only one variety (Figure 3C), which suggests that transposition may have occurred *in agro*, after domestication. Moreover, these low frequency TIPs were found in all varietal groups, regardless of their geographical origin, which further suggests that transposition is triggered in various agro-environments. However, the 32 families did not contribute to genome diversification in the same fashion (Figure 3B): some families (e.g., *Houba*) exhibit only very low frequency insertions (L-shaped distribution of the number of accessions sharing the insertions), which suggests a recent transpositional activity or a segregation of ancient polymorphisms via lineage sorting. However, some, like *Hopi*, exhibited insertions at frequencies ranging from ~ 0.001 (insertions found in only one variety) to ~ 1 (ancestral insertions found in all traditional varieties). This finding suggests that such families have undergone transposition continuously since domestication and that the low frequency of *Houba* insertions is likely not due to lineage sorting, but to recent activity. Finally, for a few families (e.g., *Dasheng*), most insertions were found at high frequencies, which suggests more ancient activity.”

was changed by :

« The level of polymorphism generated by the 32 retrotransposon families was assessed by using the frequency of insertions found in the 1,067 traditional varieties (Figure 3). Surprisingly, a large portion of TIPs are specific to only one variety (Figure 3C). The sensitivity of TRACKPOSON is high compared to other TE detection softwares but not 100 % for highly repeated families. We therefore may not exclude the possibility that some insertions may have been missed in some accessions, but this may certainly not change the L-shape distribution observed in figure 3C. Therefore, our results strongly suggest that transposition may have occurred *in agro*, after domestication. Moreover, these low frequency TIPs were found in all varietal groups, regardless of their geographical origin, which further suggests that transposition is triggered in various agro-environments. However, the 32 families did not contribute to genome diversification in the same fashion (Figure 3B): some families (e.g., *Houba*) exhibit only very low frequency insertions (L-shaped distribution of the number of accessions sharing the insertions), which suggests a recent transpositional activity or a segregation of ancient polymorphisms via lineage sorting. However, some, like *Hopi*, exhibited insertions at frequencies ranging from ~ 0.001 (insertions found in only one variety) to ~ 1 (ancestral insertions found in all traditional varieties). This finding suggests that such families have undergone transposition continuously since domestication and that the low frequency of *Houba* insertions is likely not due to lineage sorting, but to recent activity. Finally, for a few families (e.g., *Dasheng*), most insertions were found at high frequencies, which suggests more continuous activity. «

8. You say that “The higher TE density usually found in pericentromeric regions in plant genomes could thus result from retention of TEs in addition to an insertional bias in these regions for Gypsy elements.” I thought that no mechanism of preferential insertion into pericentromeric regions has been described so far (though, I may be wrong). In my

understanding, the distribution bias (not insertional bias) is solely due to TEs being preferentially removed from telomeric (or highly recombining) regions, since the removal of TEs is recombination dependent.

**But our results show that there is an insertion bias of gypsy elements in pericentromeric regions ! (cf circos plot)
How can we provide mechanistic explanation of this in the present work ?**

9. *In the sentence “These TIPs were then classified into seven distinct categories : 1), 2) and 3) : Indica, Japonica and Aus/Boro specific insertions respectively, i.e. present in at most 10 % of the other varietal groups [present in all traditional varieties, regardless their varietal group]; 4)..” – I think the part I put in the brackets has been pasted there by mistake.*
10. *TRACKPOSON pipeline – since the detection depends on the number of supporting reads – were the raw reads deduplicated prior to analyses? (e.g. by tally from the reaper program)*

Sentence was removed.

11. *I find it difficult to interpret the PcoA.*

We now use a DAPC.

12. *I suggest rephrasing at one or two places in the second-to-last paragraph of the discussion. E.g., *O. rufipogon* is not really found “all across the continent” (e.g, it is not found in Siberia), and the phrase “both sides of the Himalayas” is still confusing to me. In the next sentence, the verb “must” is not the best, since an alternative is offered a sentence later.*

Done.

REVIEWERS' COMMENTS:

Reviewer #1 (Remarks to the Author):

The description of the Nanopore data analysis is now fine.

The description of the comparative benchmark analysis has not improved. The commands used for mapping and for jitterbug have not been added. The one new info is problematic: the authors used Bowtie 2 for mapping. The default for Bowtie 2 parameters is end-to-end alignment (see <http://bowtie-bio.sourceforge.net/bowtie2/manual.shtml>). That means if the authors used default parameters Bowtie 2 does not do split-read mapping or soft-clipping. And that means the main reason to use jitterbug is absent. Hence, please either 1) show the exact bowtie command line call in the paper proving that suitable parameters were used in the benchmark, or 2) rerun bowtie with parameter --local (activating soft-clipping). Or simply use bwa-mem, which is the preferred alignment tool for this type of analysis. I would even be fine with removing the benchmark comparison completely, because the Nanopore based evaluation is strong enough by itself.

Otherwise I don't see any issues anymore.

Reviewer #3 (Remarks to the Author):

Comments to authors

Review for "Retrotranspositional landscape of Asian rice (*Oryza sativa*) revealed by 3,000 genomes"

This is a second revision. I am happy that authors now provided a time range (instead of a single time point) for the splitting the of the different rice groups.

It is still not entirely clear to me why there were three domestication events not five events.

Nevertheless I realize domestication is not the most important point of the manuscript and three or five is not really that different (both represent multiple domestication hypothesis).

I was asked to look at responses to reviewer 2's comments. Everything looks fine with me except the authors did not address question 3 from reviewer 2 at all. To me, the answer under this question is for question 2, not question 3.

I think reviewer 2 raised a valid point that there could be an element (or elements) that is absent from reference genome Nipponbare, but has generated significant polymorphisms in the population.

Practically, this question is very tough to address, particularly given the fact that the majority of the 3000 rice cultivars are not associated with an assembled genome. So it is very difficult, if not impossible, to systematically identify this type of elements.

So my suggestion is that the authors should have a brief discussion about the limitation of their study, stating that this study is based on elements mined from the reference genome...Future study will reveal whether there is exceptional activity of retrotransposons from other rice genomes. Something like that.

Reviewer #4 (Remarks to the Author):

In my view, the manuscript is now suitable for publication.

My questions and points from the previous round of reviews were addressed in this revised version or answered in the authors' response letter.

My suggestion 4. has not been addressed, but I accept this decision of the authors.

Regarding my point 8, I did not expect the authors to provide the mechanistic explanation. I think I was misunderstood. Indeed, I can see from the Fig.2 that the Gypsy retroelements are more frequent in the pericentromeric regions. However, this distribution bias can have two causes - preferential insertions into pericentromeric regions, or preferential removal from the telomeric regions. My point is that the authors do not know which of these two is the cause of the distribution bias (and a mechanism of the putative insertional bias is difficult to conceive, given the mode of LTR-retrotransposon replication). But, I understand that such discussion is beyond the scope of this manuscript.

Finally, I don't want to go into nitpicking, but my last point regarding the geography of *O. rufipogon* is still not entirely correct. The geographic term "Southeast Asia" in fact does not include the Indian subcontinent (referred to as "South Asia") or China (belonging to "East Asia"). Currently, *O. rufipogon* grows naturally from the states of western India all the way to south China, but also down to New Guinea (i.e. South Asia, Southeast Asia and parts of East Asia).

--

Regarding the points raised by the reviewer #2, I think all of them were answered satisfactorily by the authors.

In respect to the point 8, I think the authors used the best existing molecular clock estimate for LTR-retrotransposons. However, the numbering of the references has an error (two entries are numbered as 49), causing a frameshift.

In respect to the point 9, I would like to say that none of the two scenarios is the most parsimonious one. In fact, the most parsimonious scenario does not involve any introgressions, or replacement of several domestication loci, but assumes that the "domestication alleles" were present in the standing variation of various *O. rufipogon* populations, and identical variants were selected multiple times in multiple domestications. This is supported by the observation that "domestication alleles" do not cause the "domestication phenotype" in wild rice due to epistatic interactions (Inoue et al. 2015; Ishikawa et al. 2017) and are widespread in wild populations (Civan & Brown 2017). Given these prerequisites, the no-introgression scenario is the most parsimonious and not surprising (considering convergent forces in crop domestication).

P. Civan, 18.10.2018

REVIEWERS' COMMENTS:

Reviewer #1 (Remarks to the Author):

The description of the Nanopore data analysis is now fine.

The description of the comparative benchmark analysis has not improved. The commands used for mapping and for jitterbug have not been added. The one new info is problematic: the authors used Bowtie 2 for mapping. The default for Bowtie 2 parameters is end-to-end alignment (see <http://bowtie-bio.sourceforge.net/bowtie2/manual.shtml>). That means if the authors used default parameters Bowtie 2 does not do split-read mapping or soft-clipping. And that means the main reason to use jitterbug is absent. Hence, please either 1) show the exact bowtie command line call in the paper proving that suitable parameters were used in the benchmark, or 2) rerun bowtie with parameter --local (activating soft-clipping). Or simply use bwa-mem, which is the preferred alignment tool for this type of analysis.

We have used the « --end-to-end --very-fast » parameters which are in agreement with the recommendation of Jitterbug software.

I would even be fine with removing the benchmark comparison completely, because the Nanopore based evaluation is strong enough by itself.

We agree with reviewer 1 that the Nanopore-based evaluation is strong enough. In fact we propose to remove the paragraph. We ask the editors to judge whether this is ok. The paragraph is left in the latest version but strikethrough and highlighted in yellow.

Otherwise I don't see any issues anymore.

Reviewer #3 (Remarks to the Author):

Comments to authors

Review for "Retrotranspositional landscape of Asian rice (Oryza sativa) revealed by 3,000 genomes"

This is a second revision. I am happy that authors now provided a time range (instead of a single time point) for the splitting the of the different rice groups.

It is still not entirely clear to me why there were three domestication events not five events. Nevertheless I realize domestication is not the most important point of the manuscript and three or five is not really that different (both represent multiple domestication hypothesis).

I was asked to look at responses to reviewer 2's comments. Everything looks fine with me except the authors did not address question 3 from reviewer 2 at all. To me, the answer under this question is for question 2, not question 3.

I think reviewer 2 raised a valid point that there could be an element (or elements) that is absent from reference genome Nipponbare, but has generated significant polymorphisms in the population.

Practically, this question is very tough to address, particularly given the fact that the majority of the 3000 rice cultivars are not associated with an assembled genome. So it is very difficult, if not impossible, to systematically identify this type of elements.

So my suggestion is that the authors should have a brief discussion about the limitation of their study, stating that this study is based on elements mined from the reference genome...Future study will reveal whether there is exceptional activity of retrotransposons from other rice genomes. Something like that.

We added one sentence in the paragraph in order to address this point :

“We chose these three families because they are representative of the diversity of the retrotransposons found in the reference rice genome, *i.e.* a low (*Tos17*) and a high (*Hopi*) copy number LTR-retrotransposon families and a moderately repeated LINE family (*Karma*). We should however emphasize that TRACKPOSON could only unravel the transpositional activity of TE families identified from the reference genome sequence and therefore that we could not test the efficiency of this software on a retrotransposon family with genomic features that are very different from *Hopi*, *Tos17* or *Karma*, in case such family exists in one of the 3,000 rice genomes represented in the dataset. TRACKPOSON identified 501, 8 and 9 insertions of *Hopi*, *Tos17* and *Karma*, in Mutha Samba cultivar, respectively.”

Reviewer #4 (Remarks to the Author):

In my view, the manuscript is now suitable for publication.

My questions and points from the previous round of reviews were addressed in this revised version or answered in the authors' response letter.

My suggestion 4. has not been addressed, but I accept this decision of the authors.

Regarding my point 8, I did not expect the authors to provide the mechanistic explanation. I think I was misunderstood. Indeed, I can see from the Fig.2 that the Gypsy retroelements are more frequent in the pericentromeric regions. However, this distribution bias can have two causes - preferential insertions into pericentromeric regions, or preferential removal from the telomeric regions. My point is that the authors do not know which of these two is the cause of the distribution bias (and a mechanism of the putative insertional bias is difficult to conceive, given the mode of LTR-retrotransposon replication). But, I understand that such discussion is beyond the scope of this manuscript.

*Finally, I don't want to go into nitpicking, but my last point regarding the geography of *O. rufipogon* is still not entirely correct. The geographic term "Southeast Asia" in fact does not include the Indian subcontinent (referred to as "South Asia") or China (belonging to "East Asia"). Currently, *O. rufipogon* grows naturally from the states of western India all the way to south China, but also down to New Guinea (*i.e.* South Asia, Southeast Asia and parts of East Asia).*

We have modified the text accordingly :

“The distribution of *O. rufipogon* in Asia today does not contradict this hypothesis because the species is now found **all across South Asia, South-East Asia and East Asia.**”

--

*Regarding the points raised by the reviewer #2, I think all of them were answered satisfactorily by the authors. In respect to the point 8, I think the authors used the best existing molecular clock estimate for LTR-retrotransposons. However, the numbering of the references has an error (two entries are numbered as 49), causing a frameshift. In respect to the point 9, I would like to say that none of the two scenarios is the most parsimonious one. In fact, the most parsimonious scenario does not involve any introgressions, or replacement of several domestication loci, but assumes that the "domestication alleles" were present in the standing variation of various *O. rufipogon* populations, and identical variants were selected multiple times in multiple domestications. This is supported by the observation that "domestication alleles" do not cause the "domestication phenotype" in wild rice due to epistatic interactions (Inoue et al. 2015; Ishikawa et al. 2017) and are widespread in wild populations (Civan & Brown 2017). Given these prerequisites, the no-introgression scenario is the most parsimonious and not surprising (considering convergent forces in crop domestication).*

P. Civan, 18.10.2018